

# Continuous Non-Marine Inputs of Per- and Polyfluoroalkyl Substances to the High Arctic: A Multi-Decadal Temporal Record

Heidi M. Pickard[1], Alison S. Criscitiello[2], Christine Spencer[3], Martin J. Sharp[2], Derek C. G. Muir[3], Amila O. De Silva[3], Cora J. Young[1a]

[1]Department of Chemistry, Memorial University, St. John's, NL, A1B 3X7, Canada
[2]Department of Earth and Atmospheric Sciences, University of Alberta, Edmonton, AB, T6G 2E3, Canada
[3]Aquatic Contaminants Research Division, Environment and Climate Change Canada, Burlington, ON, L7S 1A1, Canada
[a]Now at: Department of Chemistry, York University, Toronto, ON, M3J 1P3, Canada

*Correspondence to*: Cora J. Young (youngcj@yorku.ca) and Amila O. De Silva (amila.desilva@canada.ca)

**Abstract.** Perfluoroalkyl acids (PFAAs) are persistent, bioaccumulative compounds found ubiquitously within the environment. They can be formed from the atmospheric oxidation of volatile precursor compounds and undergo long-range transport through the atmosphere and ocean to remote locations. Ice caps preserve a temporal record of PFAA deposition making them useful in studying the atmospheric trends in LRT of PFAAs as well as understanding major pollutant sources and production changes over time. A 15 m ice core representing 38 years of deposition (1977-2015) was collected from the

Devon Ice Cap in Nunavut, providing us with the first multi-decadal temporal ice record in PFAA deposition to the Arctic. Ice core samples were concentrated using solid phase extraction and analyzed by liquid and ion chromatography methods. Both perfluoroalkyl carboxylic acids (PFCAs) and perfluoroalkyl sulfonic acids (PFSAs) were detected in the samples, with fluxes ranging from <LOD to 141 ng m$^{-2}$ yr$^{-1}$. Our results demonstrate that the PFCAs and perflurooctane sulfonate (PFOS) have continuous and increasing deposition on Devon Ice Cap, despite recent North American regulations and phase-outs. We

propose that this is the result of on-going emission and use of these compounds, their precursors and other newly unidentified compounds in regions outside of North America. By modelling air mass transport densities, and comparing temporal trends in deposition with production changes of possible sources, we find that Eurasian sources, particularly from Continental Asia are large contributors to the global pollutants impacting Devon Ice Cap. Comparison of PFAAs to their precursors and correlations of PFCA pairs showed that deposition of PFAAs is dominated by atmospheric formation from

volatile precursor sources. Major ion analysis confirmed that marine aerosol inputs are unimportant to the long-range transport mechanisms of these compounds. Assessments of deposition, homologue profiles, ion tracers, air mass transport models, and production and regulation trends allow us to characterize the PFAA depositional profile on the Devon Ice Cap and further understand the LRT mechanisms of these persistent pollutants.



## 1 Introduction

Per- and polyfluoroalkyl substances (PFAS) are a diverse group of compounds that have been used in surfactants and polymers for over 60 years (Buck et al., 2011). Perfluoroalkyl acids (PFAAs) are persistent contaminants that are ubiquitous in the environment. Perfluoroalkyl carboxylic acids (PFCAs) and perfluoroalkyl sulfonic acids (PFSAs) are two of the most

widely known and studied groups of PFAAs (Buck et al., 2011; Stock et al., 2007). PFAAs are prevalent in remote locations, such as the Arctic (Butt et al., 2010), due to their ability to undergo long-range transport through the atmosphere and/or the ocean (Prevedouros et al., 2006). Long-range transport can be a combination of both direct transport and indirect formation. With direct transport, PFAAs are directly transported in their carboxylic (PFCA) or sulfonic (PFSA) acid form to remote locations. This can occur via oceanic water currents or by marine aerosol formation (Benskin et al., 2012b). With indirect

formation, PFAAs are produced through chemical transformation of PFAS precursors in the atmosphere (Young and Mabury, 2010). These compounds are environmentally persistent and longer chain acids (>6 carbons) have a tendency to bioaccumulate and biomagnify in food webs (Butt et al., 2010; Houde et al., 2006; Scheringer et al., 2014).

In the atmosphere, volatile and semi-volatile precursors such as fluorotelomer alcohols (FTOHs), N-alkyl perfluoroalkane sulfonamides/sulfonamidoethanols (NAFSAs/NAFSEs) and heat transfer fluids (i.e. chlorofluorocarbon-replacements)

undergo oxidation in the gas phase to form PFAAs (D'eon et al., 2006; Ellis et al., 2004; Young and Mabury, 2010). The atmospheric lifetime and persistence of these precursors is long enough to reach remote locations by wind and air transport, before subsequently oxidizing to the corresponding PFAAs and depositing to remote locations (Busch et al., 2010; Young et al., 2007).

Once these PFAAs are formed indirectly in the atmosphere, they will undergo wet or dry deposition. Further transport can

occur via ocean currents (Armitage et al., 2006, 2009a, 2009c) and marine aerosols (McMurdo et al., 2008). PFAAs are highly acidic, surface-active compounds, usually present as anions in the aqueous phase under environmental conditions (Cheng et al., 2009). These surface-active compounds will concentrate at the air-water interface and are therefore expected to be in the sea surface microlayer (SSML) and to be present in marine aerosols (Lewis and Schwartz, 2004).

The long-range transport mechanisms of these compounds can be elucidated through the collection and analysis of remote

samples, such as ice core samples. Ice caps receive their contamination solely from atmospheric deposition due to their high elevation, and preserve a temporal record of that deposition. Devon Ice Cap, located on the Devon Island in Nunavut, Canada, was previously sampled for PFAAs in May of both 2006 and 2008 through collection from the sidewall of a snow pit (MacInnis et al., 2017; Young et al., 2007). This ice cap has a high latitude and elevation (Boon et al., 2010) and is not expected to receive any local or oceanic sources of contamination. These previous studies detected PFAAs in snow profiles

that spanned a 10 – 14 year period in deposition. In this study, a 15 m ice core was collected in 2015, allowing us to examine PFAA deposition over a much longer (38 year) period. Within this paper we discuss: (1) PFAA deposition and temporal trends; (2) homologue patterns and volatile precursor mechanisms; (3) transport of PFAAs to the Arctic via ion tracer



analysis; and (4) PFAA source regions via transport modelling. This work represents the first multi-decadal analysis of PFAAs in an ice core from the summit region (2175 m above mean sea level) of a large Arctic ice cap.

## 2 Methods

### 2.1 Sample Collection and Sectioning

A 15.5 m ice core was collected from the Devon Ice Cap, Devon Island, Nunavut ($75.2°N$, $82.7°W$, 2175 m above mean sea level (AMSL)) on May 17, 2015. Samples were collected using a stainless steel Kovacs ice drill with a 9 cm diameter (section S1). The samples were separated into 1 m ice core sections, packaged in polyethylene wrap, shipped frozen to the Canada Centre for Inland Waters (CCIW) in Burlington, Ontario, Canada and stored at $-35°C$ prior to sectioning. Cores were sectioned in a $-10°C$ freezer into discrete samples corresponding to individual

years. Sectioning was done using stainless steel tools, cleaned with methanol (MeOH) (Omnisolv, 99.9%, EMD, ThermoFisher) before and after each sample. Ice core sections were placed into pre-cleaned 4L high-density polypropylene bottles and kept frozen at $-35°C$ prior to extraction and analysis.

### 2.2 Sample Preparation and Extraction

Sample extraction has been described previously (MacInnis et al., 2017). Briefly, samples were thawed immediately prior to

extraction and aliquoted into 500 mL melted volumes for extraction. Sub-samples for extraction were spiked with 30 μL of a surrogate mixture (Table S1), which acted as the internal standard (IS) to monitor recovery (Table S2). Samples were shaken, sonicated for 10 minutes, and held for 30 minutes at room temperature.

Targeted analytes included: trifluoroacetic acid (TFA), perfluoropropionic acid (PFPrA), perfluorobutanoic acid (PFBA), perfluoropentanoic acid (PFPeA), perfluorohexanoic acid (PFHxA), perfluoroheptanoic acid (PFHpA), perfluorooctanoic

acid (PFOA), perfluorononanoic acid (PFNA), perfluorodecanoic acid (PFDA), perfluoroundecanoic acid (PFUnDA), perfluorododecanoic acid (PFDoDA), perfluorotridecanoic acid (PFTrDA), perfluorotetradecanoic acid (PFTeDA), perfluorohexadecanoic acid (PFHxDA), perfluorooctadecanoic acid (PFOcDA), perfluorobutane sulfonic acid (PFBS), perfluorohexane sulfonic acid (PFHxS), perfluoroheptane sulfonic acid (PFHpS), perfluorooctane sulfonic acid (PFOS), perfluorodecane sulfonic acid (PFDS), perfluoroethylcyclohexanesulfonate (PFECHS), and perfluorooctanesulfonamide

(FOSA).

Samples were concentrated using an OASIS® weak anion exchange solid phase extraction (SPE) cartridge ($6\ cm^3$, 150 mg, 30 μm). Cartridges were conditioned prior to sample loading with 5 mL 0.1% $NH_4OH$/MeOH, followed by 5 mL MeOH and 5 mL SPE-cleaned HPLC Grade water (Fisher). Following sample concentration, the cartridges were rinsed with 25 mM ammonium acetate buffer acidified to pH 4 with acetic acid, and centrifuged at 4000 rpm for 2 minutes to remove any

residual water. Samples were eluted into two fractions: the first fraction was eluted with 6 mL of MeOH for FOSA, and the second fraction was eluted with 8 mL of 0.1% $NH_4OH$/MeOH for PFAAs. Both fractions were evaporated to dryness under a



gentle stream of nitrogen and reconstituted in 0.5 mL 50/50 methanol – water containing the surrogate mixture (Table S1) to monitor matrix effects (Table S3). Reconstituted samples were sonicated for 5 minutes, vortexed and transferred to polypropylene vials for analysis.

### 2.3 Quality Assurance/Quality Control

Cartridge blanks were used to validate the integrity of the extraction method, and isotopically labelled standards were used to validate recovery and matrix effects. Three samples were extracted in triplicate and one sample in duplicate to evaluate reproducibility. A composite mixture of Devon Ice Cap samples was prepared for three types of QA/QC measures in triplicate: one sample was spiked with PFASs before extraction, one sample spiked with PFASs after extraction, and the third sample was spiked with the internal standard and processed akin to the larger sample set. The pre-extraction and post-

extraction spiked samples were compared to evaluate recovery and matrix effects. No quantifiable PFAA levels were detected in the routinely analyzed methods blanks (SPE cartridge blanks, n=6). Method recoveries for the PFAAs ranged from 79 – 117% with the exception of PFOcDA (127 – 225%). PFOcDA recoveries indicated enhancement of analyte signal due to matrix effects and incorrect recovery and matrix correction due to using MPFHxDA as the internal standard. Since PFOcDA was below detection limit in all ice core samples, this was not explored any further.

Matrix effects were evaluated by comparing the peak area of instrument performance standard (IP) compounds to peak areas at equivalent concentrations in a solvent standard. Recovery was evaluated by comparing the recovered analyte concentration in the spike and recovery sample to the theoretical spiked concentration. Each sample was corrected for recovery and matrix effects by quantifying the concentrations based on relative response to isotopically labelled standards added before extraction. A 15 level calibration curve was employed ranging from $0.02 - 8.5$ ng mL$^{-1}$, along with analytical

blanks. Analytical blanks (MeOH) and cartridge blanks were included in the method analysis. The method detection limit (MDL) was based on three times the standard deviation of the cartridge blanks. Most PFAA analytes were not detected in the method blanks and were therefore below the instrument detection limit (IDL) (Table S4). The limit of detection (LOD) and limit of quantitation (LOQ) were quantified based on signal-to-noise (S/N) ratios of 3 and 10, respectively (Table S5).

### 2.4 Sample Analysis

Samples were analyzed by ultra performance liquid chromatography (Waters Acquity UPLC I) with tandem mass spectrometry (Waters Xevo® TQ-S, UPLC-MS/MS) detection operated in electrospray negative ionization mode. Two analyses were conducted. For the long-chain PFAA (PFCAs > C4, PFSAs > C4, FOSA) analysis, samples were separated using a C18 column (Waters Acquity UPLC® BEH, 2.1×50 mm, 1.7 μm) with a water – methanol 2 mM ammonium acetate gradient method. For the short-chain PFAA (PFCAs < C8, PFOS) analysis, samples were separated using a Shodex RSPak

column (2.0×150 mm, 5 μm) with a water – methanol 50 mM ammonium acetate method (Tables S6-S7). Analytes were quantified based on relative response to isotopically labelled internal standards (Wellington Laboratories, Guelph, ON).



### 2.5 Major Ion Analysis

Sub samples of the sectioned ice core (15 mL) were analyzed for major anions and cations. Anions were measured by ion chromatography with conductivity detection and cations were quantified using inductively coupled plasma and optical emission detection. A range of cations ($Na^+$, $K^+$, $Ca^{2+}$, $Mg^{2+}$, $Mn^{2+}$, $Al^{3+}$) and other metals (e.g. iron and silicon), and anions

($F^-$, $Cl^-$, $Br^-$, $NO_2^-$, $NO_3^-$, $SO_4^{2-}$, $PO_4^{3-}$) and organic acids (e.g. acetate, propionate, formate and butyrate) were measured. Further details on anion and cation analysis are provided in section S2.

### 2.6 Air Mass Transport Densities

To trace the origins of air masses arriving at the sampling site on Devon Ice Cap and characterize source regions, air mass back trajectories were computed using the HYbrid Single-Particle Lagrangian Integrated Trajectory (HYSPLIT) model

(Stein et al., 2015). Air parcel back trajectories were computed, allowing us to examine air mass moisture source regions and transport to Devon Ice Cap. Back-trajectory analysis was performed using the National Centers for Environmental Protection and Atmospheric Research (NCEP/NCAR) global atmospheric reanalysis data set at 2.5 degrees resolution. Daily, 10-day back trajectories were initiated at 2175 meters AMSL at the Devon Ice Cap sample site location for years 1994 and 2013. Residence time analysis was used to identify air mass transport densities for 1994 and 2013 (Miller et al., 2002). This

approach analyzes a large number of trajectories to reduce uncertainties, develop reliable pathways of airflow and account for variations in transport speed and direction (Criscitiello et al., 2016). For this analysis, the total number of trajectory endpoints were summed within each equal-area pixel, and then divided by the zonal distance between the Devon ice core sampling site and each pixel to remove concentric patterning. The air mass transport densities were scaled on a 0-1 scale. In this study we focus on low-elevation air masses (0–500 m above terrain), which are more likely to be representative of

evaporation moisture source.

### 3 Results and Discussion

### 3.1 PFAA Concentrations and Fluxes on the Devon Ice Cap: Comparisons to Previous Studies

A comprehensive analysis of perfluoroalkyl acid (PFAA) and FOSA annual deposition on the Devon Ice Cap was carried out on ice core samples dating from 1977 – 2015. In general, PFCAs from TFA (C2) to PFTrDA (C13) were detected on the

Devon Ice Cap. Deposition and temporal trends of TFA (C2) to PFBA (C4) will be discussed in a separate paper. Observed PFCA concentrations from PFPeA (C5) to PFTrDA ranged from <3.21 to 755 pg $L^{-1}$ (Fig. S1, Table S8). PFCAs from PFHxA to PFUnDA (C6 – C11) were detected in almost every sample, with the exception of PFUnDA, which was not detected in one year. The long-chain PFCAs, PFDoDA and PFTrDA were only quantifiable in <3 years, while PFTeDA, PFHxDA, and PFOcDA were <LOQ throughout the 38-year time period and will not be discussed further. PFSAs including

PFBS, PFHpS and PFOS, as well as FOSA were detected on the Devon Ice Cap with concentrations ranging from <0.18 pg





L$^{-1}$ to 391 pg L$^{-1}$ (Table S9). PFOS was detected in every sample while PFBS and PFHpS were only detected >LOD in two years. Evidence of the presence of PFHxS, PFDS and PFECHS was sought, but not found. FOSA was detected >LOD in most samples up until 2000 and only in three samples after 2000.

Concentrations (pg L$^{-1}$) of the PFAAs were converted to fluxes (ng m$^{-2}$ yr$^{-1}$) (Section S3) to determine annual deposition of
these compounds in the Canadian Arctic. Annual snow accumulation was 0.15 – 0.64 m yr$^{-1}$, which is consistent with 0.22 – 0.24 m wet equivalents yr$^{-1}$ reported by (Pinglot et al., 2003) for Devon Ice Cap from 1963-2000. Fluxes of PFAAs, including PFOA to PFUnDA (C8 – C11) and PFOS (Tables S10, S11), were compared to fluxes of these PFAAs in two previous samples collected from snow pits on Devon Ice Cap in 2006 (Young et al., 2007) and 2008 (MacInnis et al., 2017). The data sets generally agree within the uncertainty of the measurements, with the exception of PFOS (Fig. S2, S3). These
slight discrepancies between Devon Ice Cap studies could be the result of multiple factors. In the 2006 study, there was limited availability of isotopically labelled and native standards of sufficient purity. Our study represents an improvement in analytical methods since that time, both in terms of instrument detection limits and accuracy. The 2006 and 2008 sampling strategies are also in contrast to the current approach of ice core drilling. In the earlier efforts, depth samples were obtained by horizontal cylindrical sampling the face of an ice pit (2006) and vertically sampling the face of a snow pit continuously
(2008). Those datasets represent semi-continuous depth measurements. In our current method, we obtained ice cores using a custom drill designed for Arctic sampling and conventional practices in ice sampling for temporal profiling (Boon et al., 2010; Readinger, 2006). Lastly, the Devon Ice Cap sampling locations in the earlier research were not at the summit of the ice cap as in the current research. These differences in location may have resulted in some variability in fluxes. The most accurate temporal record of atmospheric deposition is obtained at the summit of an ice cap; thus, the current research is
hypothesized to be a better representation of PFAA deposition to Devon Island.

Concentrations and homologue trends of PFCAs and PFSAs detected in this study are comparable to a number of other studies (Table S12). Comparable levels of PFCAs ranging from PFPeA to PFUnDA were detected in remote snow core and surface snow samples from Cole Gnifetti (Kirchgeorg et al., 2013), Longyearbreen (Kwok et al., 2013), and from glaciers on the Tibetan Plateau and Lake Namco (Wang et al., 2014a). Short-chain PFCAs had higher concentrations than other PFCAs
detected in precipitation, lake and river water samples (Kwok et al., 2010; Müller et al., 2011; Scott et al., 2006; Taniyasu et al., 2008). In general, concentrations of the PFCAs were much lower in the Arctic and Atlantic Oceans compared to the Devon Ice Cap concentrations. PFOS and FOSA concentrations were variable among all studies with no discernable trend for sample matrix or sample site.

### 3.2 Air Mass Transport Density Analysis

Previous studies using back trajectory analysis of air masses (Kahl et al., 1997; Meyer et al., 2012) have provided evidence for atmospheric LRT to the Devon Ice Cap from North America and Eurasia. Air masses on Devon Island originated three times more often from populated regions of Northern Europe and Asia compared to North America, and southern and eastern Asia were significant source regions. Little variation was observed in Devon Ice Cap air mass source regions over the time



period 1994 – 2008 (Meyer et al., 2012). Previous studies on spatial patterns of pollen deposition in the High Arctic further support these findings. Devon Ice Cap is located within an air mass boundary between 74° N and 76° N, between sites north of 76° that receive dominantly Eurasian pollen sources and sites south of 74° that receive dominantly North American pollen sources (Bourgeois et al., 2001). Devon Ice Cap therefore receives air masses and pollen/pollutant sources from both North

America and Eurasia.

Air mass transport densities have previously been employed for investigating probable source regions and flow pathways of air masses in the Canadian Arctic (Criscitiello et al., 2016). Air mass transport density analyses calculated using the HYSPLIT model for Devon Ice Cap for the years 1994 and 2013 are shown in Figure 1. The year 1994 was selected to compare to previous trajectory analyses. Because of our unusual PFAA observations in 2013, we also examined this year to

identify any transport anomalies, but found both years (1994 and 2013) to be comparable. Air mass transport densities for both 1994 and 2013 reveal elevated residence time densities in parts of Asia, and particularly high residence time densities along the west coast of Greenland.

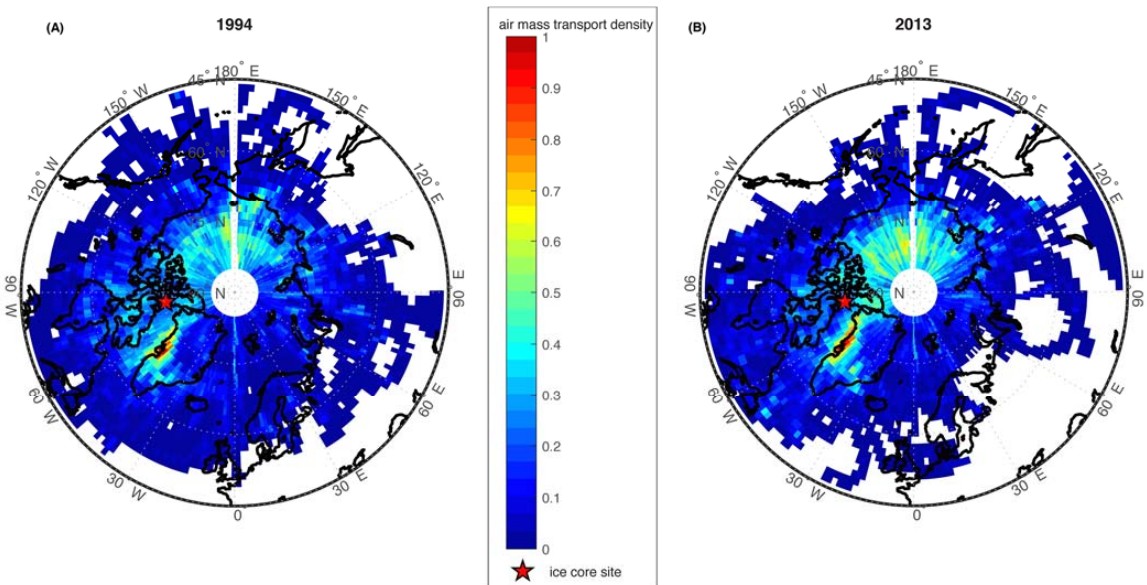

**Figure 1: Air mass transport density maps (scaled 0-1) for air parcels reaching the Devon Ice Cap ice core site (red star), for a) 1994 and b) 2013.**





### 3.3 PFCA Deposition and Temporal Trends on Devon Ice Cap

Annual fluxes of PFCAs ranged from <LOD to 141 ng m$^{-2}$ yr$^{-1}$ (Table S10). Temporal trends of short-chain PFCAs (C2 – C4) will be discussed separately, so this section will focus solely on PFCAs with more than five carbons. In general, the flux of PFCAs increased after 1985 (Fig. S4), but with diverging trends from 1995 – 2015. Various glaciology studies have

shown that Devon Ice Cap has experienced strong summer warming since 2000 and especially after 2005 (Bezeau et al., 2013; Gascon et al., 2013; Sharp et al., 2011). This is consistent with Inuit traditional knowledge of overall warming in the Arctic (Koihok et al., 2001). The variability in fluxes within the past 15-20 years could, therefore, be partially attributed to melting effects. When melting events occur, the ability to discern temporal trends in chemical deposition of various compounds can be compromised by the percolation of meltwater or elution of particles by meltwater flow (Eichler et al.,

2001; Steinlin et al., 2016). These melting events could bias annual flux measurements of PFAAs in ice core samples during the melt period, since PFAAs on the ice cap surface can be eluted into the snowpack, before refreezing at ice layer interfaces where temperatures at depth are below the pressure melting point (Bezeau et al., 2013). One study has examined elution behaviour of PFAAs from a melting snowpack and found that elution of PFAAs is largely driven by water solubility in the snowpack (Plassmann et al., 2011). This melting has likely happened periodically over the last 15 – 20 years, thereby

blurring to some extent, the vertical profile. However, we expect any melting that occurred to have primarily affected the seasonal trends. Variability of ± 1 year could be caused by inaccuracies in dating and/or error associated with ice core sectioning. To circumvent the compounding impacts of recent melt events and consequent meltwater percolation, and any error associated with ice core dating and sectioning, a 5-year moving average was applied to the flux measurements, thereby facilitating long-term temporal trend analysis of PFAS deposition to Devon Ice Cap.

Both PFOA and PFNA fluxes increased from 1977 up until at least 1995 (Figure 2). In the period post-1995 until 2013, fluxes have plateaued, with <25 ng m$^{-2}$ yr$^{-1}$ variance in annual flux. From 2012 to 2015, a large decline in PFOA and PFNA fluxes is apparent. The decrease in flux post-2012 was noted for the entire suite of PFCAs (Fig. S4). The most recent decrease in PFCA fluxes could be due to melting events or ice core dating and sectioning inaccuracies, but it also corresponds to anticipated PFCA emission reductions through the United States Environmental Protection Agency (EPA)

PFOA Stewardship Program (US EPA, 2016), as well as the Canadian Environmental Performance Agreement (ECCC, 2006). In 2006, the EPA invited eight major fluoropolymer and fluorotelomer manufacturers to commit to eliminating emissions and product content levels of PFOA, precursor compounds, and related longer chain length homologue chemicals. Corporations voluntarily committed to achieving a 95% reduction by 2010, measured from a year 2000 baseline, and full elimination of these products and emissions by 2015 (US EPA, 2016). In Canada, the federal government established The

Environmental Performance Agreement with the same commitment between Environment and Climate Change Canada (ECCC), Health Canada and four major manufacturers with known organofluorine products in Canadian commerce (ECCC, 2006). The most recent data for company-reported reductions in emission and product content for the U.S., Canada, and non-U.S. operations are summarized in Tables S13 – S14. As of 2016, all companies participating in the PFOA Stewardship



Program and Environmental Performance Agreement reported they had met the goals of the program. As part of both agreements, all major manufacturers reduced their production and emissions of PFOA and related compounds by at least 95% from 2006 – 2010. The observed decrease in PFOA and PFNA fluxes from 2012 – 2015 cannot solely be attributed to these phase-outs, since it would be expected that the phase out would cause a large decrease in PFCA deposition between

5   2006 and 2010 and a small decrease after 2012. It is probable that existing products continued to emit after the stewardship program took effect, which could delay the detection of its impact (Prevedouros et al., 2006). This is evident in temporal trend analysis in Canada and the U.S. such as in human blood, freshwater fish, and non-migratory birds, which do not show any declines in PFCAs from 1990s to 2012 (Braune and Letcher, 2013; Gewurtz et al., 2016).

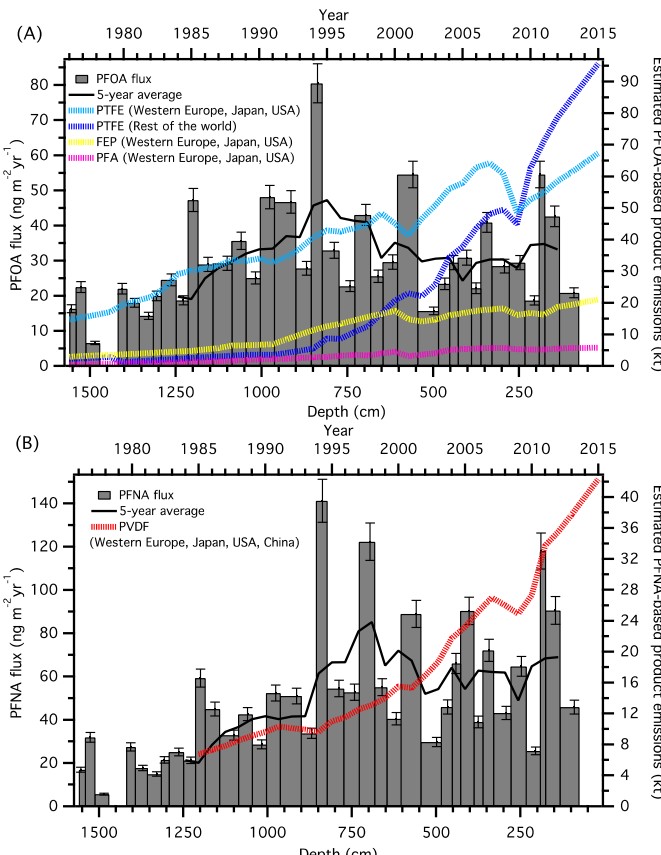

10   **Figure 2: Annual deposition fluxes on the Devon Ice Cap: (a) PFOA and (b) PFNA. The solid black line represents the 5-year moving average and the dotted coloured lines represent the estimated and reported consumption or production volumes of PFOA-based products including PTFE, perfluorinated ethylene-propylene copolymers (FEP), perfluoroalkoxyl polymers (PFA), and related ammonium and sodium salts (APFO/NaPFO), as well as PFNA-based products including polyvinylidene fluoride (PVDF) and related ammonium salts (APFN) (Wang et al., 2014b).**



In addition, other historical and on-going sources of PFAS have emerged. Manufacturers that were not signatories to the PFOA stewardship program have been producing PFAS since 1985. China started polytetrafluoroethylene (PTFE) production around 1985 and was producing up to 60 kt of PTFE in 2015 (Wang et al., 2014b). Some manufacturers have

emerged more recently that produce PFAAs and precursor compounds (Land et al., 2015; Wang et al., 2014b). In general, from 2000 onward, long-chain PFCAs have been phased-out through regulation or voluntary reduction by major producers in Japan, Western Europe, the United States and Canada (Wang et al., 2014b, 2014c). Meanwhile, new manufacturers (e.g. China) have begun producing these long-chain PFCAs and their precursors (Land et al., 2015). The total estimated annual emissions of PFBA to PFTeDA for Canada, the United States, Western Europe, and Japan were $25 - 50 \, t \, yr^{-1}$ in 2010, while

estimated emissions of PFBA to PFTeDA were $40 - 193 \, t \, yr^{-1}$ in China in 2013 (Wang et al., 2014b). Thus, global emissions of PFAAs and their precursors have not decreased significantly, which is consistent with observed temporal trends on Devon Ice Cap. Furthermore, the phase-out has created a market for numerous alternative fluorinated products (Wang et al., 2016, 2013), some of which may be precursors to long-chain PFCAs. Observed temporal trends in PFAA fluxes may represent combined-effects of the stewardship program in North America and increasing production and emissions in other regions.

**3.4 PFSA and FOSA Deposition and Temporal Trends on Devon Ice Cap**

The observed concentrations of PFSAs and FOSA correspond to annual fluxes from $<$LOD to $80.3 \, ng \, m^{-2} \, yr^{-1}$ (Table S11). Since PFBS and PFHpS were only detected in three samples, there are no observable trends. PFOS and FOSA each show distinct temporal trends. PFOS was detected at consistent levels below $10 \, ng \, m^{-2} \, yr^{-1}$, with an anomaly ($80 \, ng \, m^{-2} \, yr^{-1}$) detected in 2012 (Fig. 3a). FOSA was measured in almost every year from 1977 – 2000 with fluxes increasing until 1995.

After 2000, FOSA was only detected in three samples with levels $<0.76 \, ng \, m^{-2} \, yr^{-1}$ (Fig. 3b). FOSA is a known volatile precursor and can degrade to PFOS (D'eon et al., 2006); however, PFOS was continually measured after 2000, whereas FOSA was not. In addition, there was no correlation between PFOS and FOSA measurements (Table S15). This suggests FOSA is not the primary source of PFOS to the Devon Ice Cap.





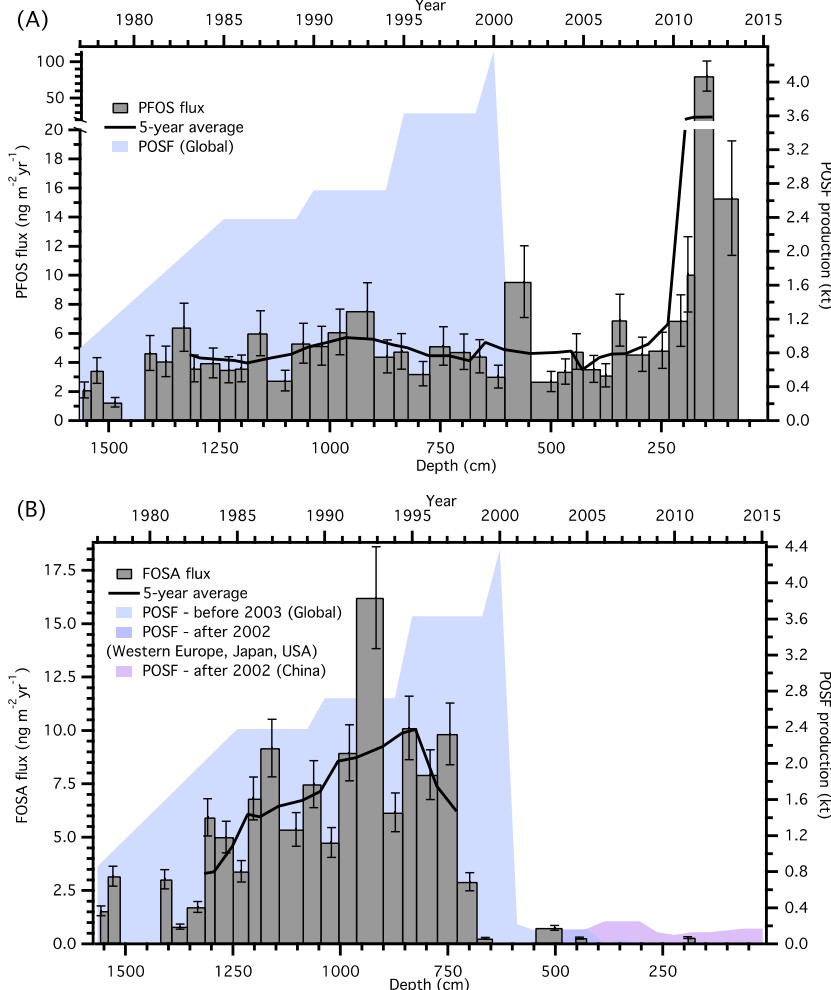

**Figure 3: Annual deposition fluxes on the Devon Ice Cap: (a) PFOS and (b) FOSA, with global POSF production from 1977 – 2003 and POSF production for Western Europe, Japan, United States and China from 2003 – 2015 (Wang et al., 2014b, 2017). Solid black lines represent 5-year moving averages.**

From 2000 – 2002, 3M, one of the major global producers of perfluoroalkyl substances phased out the production of the synthetic precursor to FOSA and PFOS, perfluorooctane sulfonyl fluoride (POSF), as well as related products based on C6, C8, and C10 chemistry (Wang et al., 2013; Weppner, 2000). These were replaced with C4-based chemistry, where products were derived from perfluorobutane sulfonyl fluoride (PBSF). These compounds are believed to have lower bioaccumulative



and toxicological effects (Stahl et al., 2011). Reported production of PBSF in the United States almost doubled from 2002 to 2006, while that of POSF decreased by more than two orders of magnitude between 1998 and 2002, with no known production after 2006 (Wang et al., 2014c). Before 2003, 3M was responsible for most global PFOS production (Carloni, 2009). By 2006, PFOS and related substances derived from POSF were regulated under the European Union (EU) Directive

2006/122/EC and by 2009, were listed under Annex B (restriction of production and use) of the Stockholm Convention on Persistent Organic Pollutants coordinated by the United Nations Environment Programme (UNEP) (Land et al., 2015). These production changes in PFOS in the early 2000s were used to explain temporal trends in the Canadian Arctic (Butt et al., 2007). The decline in production of the precursor FOSA by 3M is consistent with FOSA trends observed on the Devon Ice Cap, in which the majority of FOSA is <LOD after 2000, suggesting an effective phase out of this compound (Fig. 3b). This

was similarly observed in samples collected from Devon Ice Cap in 2008 (MacInnis et al., 2017), in North Atlantic pilot whales harvested between 1986 and 2013 (Dassuncao et al., 2017), and in Arctic air at Alert (Hung et al., 2016).

Temporal trends of PFOS deposition to the Devon Ice Cap do not reflect the production phase out of PFOS by 3M in the early 2000s (Fig. 3a). Rather, PFOS production and manufacturing, along with replacements (e.g. PFBS), have increased dramatically in Asia since 2001, and China is now the dominant producer of these compounds (Lam et al., 2016; Xie et al.,

2013; Yao et al., 2016). PFOS production in China began increasing rapidly around 2000 and is currently steady at 100 – 200 t $yr^{-1}$ (Wang et al., 2016). China was reported to be the main producer and user of PFOS substances between 2003 – 2008 with less than 50 t in 2003 and up to 250 t of POSF-based products produced in 2008 (Carloni, 2009). During this time, over 100 t of PFOS was also used annually in China to produce aqueous film forming foams (AFFFs) (Armitage et al., 2009b) used for extinguishing fuel-based fires. According to available 2006 inventories, 15 Chinese enterprises were

producing over 200 t of POSF, of which 100 t were for export (Ruisheng, 2008). This suggests the annual volume of PFOS production in China in the mid-2000s was similar to the annual production by 3M in the late 1990s (Armitage et al., 2009b). Although it is known that the production of perfluoroalkane sulfonyl fluorides has increased in China, global emission data for individual compounds are currently unavailable and cannot be correlated with the temporal trends observed on the Devon Ice Cap (Lim et al., 2011). However, the continuous detection of PFOS after the early 2000s on Devon Ice Cap may be

related to the on-going production and use of PFOS substances by manufacturers in Asia (Wang et al., 2017). Production of other PFOS-related perfluorinated chemicals is on-going in China, as well as in Russia and India (Jiang et al., 2015) which is supported by higher levels of PFOS after 2011 in the Devon Ice Cap. An anomalously high PFOS flux was observed in 2013. This flux was between five and eight times greater than both previous and following years. There was no signal enhancement for other PFAS in 2013 and the air mass transport model showed no transport anomaly for this year. Therefore,

the large 2013 PFOS flux in Devon Ice Cap is unlikely attributed to contamination during method collection or analysis, or due to air mass movements. The reason for the high 2013 PFOS flux is unknown but may suggest other sources. Arctic air samples collected at Alert also showed the highest levels of PFOS in 2013 (Hung et al., 2016), and anomalously high PFOS levels were observed in landlocked Arctic Char from Cape Bounty, Melville Island in Nunavut, Canada, collected between





2011 – 2015 (Cabrerizo et al., 2016). This increasing PFOS trend warrants further consideration in order to determine the efficacy of current POSF restrictions.

**3.5 PFCA Homologues and Volatile Precursors**

Indirect sources of PFAAs are contributors to the global presence of these compounds, but contamination is more important
for certain homologues in some locations, for example the Canadian Archipelago (Benskin et al., 2012a). This has been demonstrated in multiple studies that detected the presence of volatile precursors (e.g. FTOHs, NAFSAs, NAFSEs), and FTOH-precursor degradation products (e.g. fluorotelomer unsaturated carboxylic acids (FTUCAs)), in the Canadian Arctic (Benskin et al., 2011; Schenker et al., 2008; Shoeib et al., 2006; Stock et al., 2007). Volatile precursor compounds will oxidize in the atmosphere to produce PFCAs. Patterns of PFCA homologues are useful in examining the role that
fluorotelomer-derived compounds play in gas-phase atmospheric oxidation (Young et al., 2007). If these compounds are coming from the same source, then sequential pair concentrations are expected to vary through time together. There will be some variability in the ratios depending on the relative atmospheric levels of $NO_x$ ($NO + NO_2$) and peroxy radicals (Young and Mabury, 2010). In this study, comparisons were made between observed concentrations of 36 pairs of PFCA homologues (Table S15). Most sequential pairs of PFCA homologues were deposited in similar amounts on the ice cap.
Correlations between sequential pairs from PFPeA and PFDoDA were all statistically significant (two-tailed t-test) with strong correlations (all p-values $\leq$ 0.0001; $0.527 \leq R^2 \leq 0.889$; Table S15). Similar correlations were observed for a number of PFCA homologues in previous ice samples from the Devon Ice Cap (MacInnis et al., 2017) and the Longyearbreen glacier in Svalbard (Kwok et al., 2013). The correlations are consistent with expected PFCA homologue production via gas phase atmospheric oxidation of fluorotelomer-derived compounds (Ellis et al., 2004). Flux measurement ratios were calculated for
six pairs of PFCA homologues from PFPeA to PFUnDA over the time series (Fig. S5). The three major even-odd pairs expected to be formed from 6:2, 8:2, and 10:2 fluorotelomer compounds are PFHxA:PFHpA, PFOA:PFNA, and PFDA:PFUnDA, respectively. The majority (82%) of the flux ratio measurements were within a factor of two, supporting the hypothesis that these sequential, even-odd homologues are likely coming from fluorotelomer-derived sources (Fig. 4a) (Ellis et al., 2004; Wallington et al., 2006). Fluorotelomer compounds of different chain lengths were produced and used to
different extents. We can compare these using odd-odd PFCA homologue ratios, comparing PFNA to PFPeA, PFHpA, and PFUnDA, as products of 8:2, 4:2, 6:2, and 10:2 fluorotelomer compounds, respectively (Fig. 4b). The dominant homologues are PFNA and PFHpA, followed by PFPeA, then PFUnDA. This suggests that 8:2 and 6:2 fluorotelomer compounds dominate as precursors, followed by the 4:2 and 10:2 fluorotelomer compounds. This trend is consistent with our knowledge of commercial product formulations and atmospheric measurements (Dinglasan-Panlilio and Mabury, 2006; Heydebreck et
al., 2016; Young and Mabury, 2010). Despite producers moving from 8:2 to shorter-chain formulations, recent FTOH atmospheric measurements have found that the 8:2 FTOH remains the dominant compound in the High European Alps (Xu et al., 2017). Therefore, it is likely that PFCAs from PFPeA to PFUnDA on the Devon Ice Cap are derived from common emission sources due to prominent quantities of residual volatile precursors in fluoropolymer products.





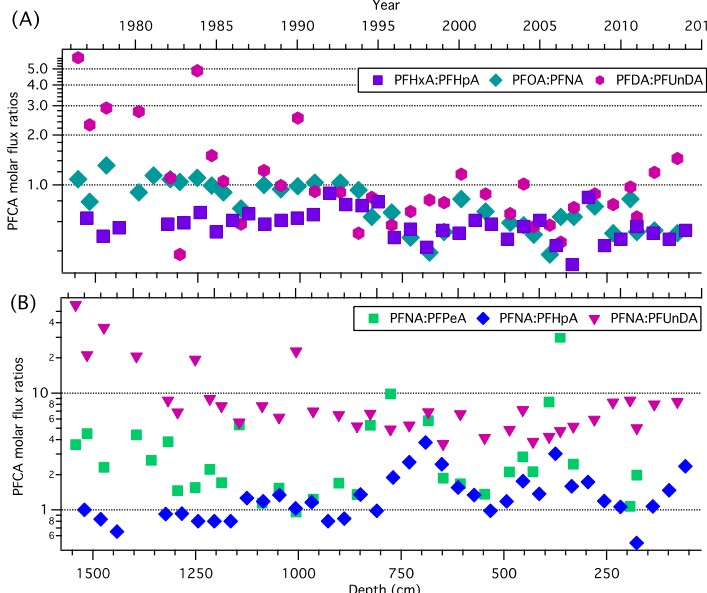

**Figure 4: Molar flux ratios for (a) three even-odd pairs of PFCAs and (b) three odd pairs of PFNA with PFPeA, PFHpA, and PFUnDA homologues, as a function of depth and year.**

The atmospheric oxidation of perfluoroalkane sulfonamido substances may provide an additional source of PFCAs to the Devon Ice Cap. The oxidation of FOSA could contribute to the observed flux of PFOA and shorter-chain PFCAs in the High Arctic (Martin et al., 2006). However, since there were no observed correlations between FOSA and PFOA or FOSA and any of the other PFCAs ($p \geq 0.0021$; $R^2 \leq 0.24$, Table S15), it is likely that PFOA deposition to the Devon Ice Cap is driven

10 by other sources.

### 3.6 Elucidating the Role of Marine-Driven Transport of PFAAs to the Arctic

Thus far, evidence was provided for indirect formation of PFAAs in the atmosphere from volatile precursor compounds, indicating that direct transport is an unlikely source of PFAAs to the ice cap. Atmospherically formed PFAAs can deposit to any terrestrial or oceanic system, and to the atmosphere into the ocean before reaching the Arctic, and then re-enter the

15 atmosphere from oceanic currents and marine aerosols. It is therefore important to understand the extent to which the oceans contribute to PFAA ice cap deposition. Other sources, such as dust or biomass burning, must also be considered as possible contributors of PFAAs to the ice cap. Gas-phase PFAAs will deposit through wet or dry deposition and may also be





transported on aerosols (Thackray and Selin, 2016). Major ions are useful source markers for atmospheric aerosols, and were measured in the ice core to further understand these transport mechanisms (Tables S16 – S17, Fig. S6 – S7).

The non-sea salt (nss) component of the ice core samples was calculated to understand the atmospheric origin in the samples (Keene et al., 1986). All sodium (Na$^+$) in the ice core samples was assumed to come from sea salt. Most other ions were

attributed to the nss component, suggesting limited oceanic sources depositing on Devon Ice Cap (Table S18). Further, no correlations were observed between Na$^+$ flux and any of the PFAAs (p $\geq$ 0.0093; R$^2$ $\leq$ 0.218; Table S19).

Another technique used to assess the influence of marine aerosol deposition of PFAAs to the Devon Ice Cap is a comparison between ocean and ice cap homologue patterns. If marine aerosols were/are a major source of PFAA contamination on Devon Ice Cap, then one would expect the homologue profiles to be similar between the ocean and the ice cap. Figure 5

illustrates the proportional analysis of the molar concentration (pmol L$^{-1}$) fraction between PFAAs on the Devon Ice Cap and ocean levels in the Canadian Artic Archipelago (Benskin et al., 2012b), Arctic Ocean (Benskin et al., 2012b; Cai et al., 2012a), North Atlantic Ocean  (Zhao et al., 2012), and North Pacific Ocean (Cai et al., 2012a) for the years 2005 and 2010. Molar concentrations of PFAAs on the ice cap differed from ocean PFAA concentrations, with higher molar concentrations of PFBA and PFNA, and lower molar concentrations of PFSAs found on Devon Ice Cap. PFHxS was not detected on the ice

cap, but was measured in most ocean samples, suggesting that indirect sources are of importance for Arctic deposition, due to the absence of PFHxS on the ice cap. The differences in homologue profiles between the ice cap and the ocean can also not be accounted for by different surfactant properties (MacInnis et al., 2017), suggesting the two may have different sources of PFAA contamination. The discrepancies between the Na$^+$/PFAA flux ratios and the ice cap/ocean proportional analysis provide further evidence to imply that marine aerosols are not a significant source of PFAAs to the Devon Ice Cap.

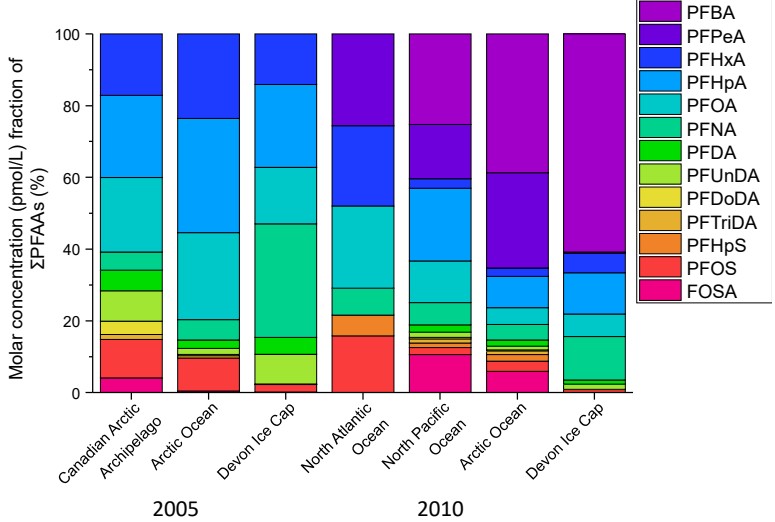





**Figure 5: Molar concentration fraction of sum of PFAAs on the Devon Ice Cap compared to levels in the Canadian Arctic Archipelago (Benskin et al., 2012b), Arctic Ocean (Benskin et al., 2012b; Cai et al., 2012a), North Atlantic Ocean (Zhao et al., 2012), and North Pacific Ocean (Cai et al., 2012a) in 2005 and 2010. Ocean concentrations are compared to ice cap concentrations for two years to show a better overall comparison with different ocean samples that were collected in multiple field campaigns.**

**3.7 Understanding Atmospheric Inputs of PFAAs Using Ion Tracers**

Weak correlations were observed between short-chain PFCAs and nss-F$^-$ (p $\leq$ 0.0015; 0.306 $\leq$ R$^2$ $\leq$ 0.455). A small percentage of the F$^-$ being detected in the ice core samples may be derived from the atmospheric formation of PFCAs. The degradation of many PFCA precursors, including heat transfer fluids, forms both PFCA and HF. For example, the hydrolysis of perfluoroacyl fluorides forms the corresponding PFCAs along with the loss of HF (Calvert et al., 2008):

$CF_3(CF_2)_xCOF + H_2O \rightarrow CF_3(CF_2)_xCOOH + HF$ (1)

By this mechanism in Eq. (1), PFCAs could account for between 0.80 – 14% of the F$^-$ present on the ice cap, depending on the year. These numbers are upper limits as PFCAs can also be formed by mechanisms that do not form HF (Young and Mabury, 2010). It is difficult to assess the exact contribution of this anthropogenic source to the overall burden of F$^-$, due to the lack of available data on F$^-$ sources. This is further confounded by the high mobility of F$^-$ in both firn and ice layers,

which makes it difficult to study temporal trends in F$^-$ deposition (Preunkert et al., 2001). Both natural and anthropogenic sources contribute to the overall budget of F$^-$ in the troposphere, including primary sea-salt, soil dust aerosols, volcanic emissions, coal burning, and industrial processing (Preunkert et al., 2001). We are currently unable to assess the exact contributions of each source of F$^-$, but the correlations observed here suggest that short-chain PFCA precursor degradation could account for up to 14% of the observed F$^-$. Furthermore, these correlations lend further support to indirect formation as

a major pathway to PFCA contamination on the Devon Ice Cap.

There were weak to moderate correlations between several PFAAs and nss-Ca$^{2+}$ and nss-Mg$^{2+}$ (0.300 $\leq$ R$^2$ $\leq$ 0.531), both of which are indicators of mineral dust (Mochizuki et al., 2016). Mineral dust aerosols can accumulate acidic atmospheric contaminants, such as nitric and hydrochloric acids, and undergo LRT during dust storms, where they can travel long distances from Asian and African dust sources to the remote Arctic (Rahn et al., 1977; Sullivan et al., 2007). Dust

entrainment in deserts is one of the most important sources of mineral dust in the global atmosphere and North African (e.g. Sahara) and Central Asian (e.g. Gobi desert) dust sources contribute the most global dust to the Northern Hemisphere (Luo et al., 2003). Several studies have found dust particles associated with LRT in snow and ice samples from the Canadian Arctic (Bullard et al., 2016; Groot Zwaaftink et al., 2016; Zdanowicz et al., 1998). Dust deposition to the Arctic shows a seasonal effect, with dust storms in major deserts occurring more frequently in the spring, leading to higher concentrations of

mineral dust tracers in the spring and autumn (Barrie and Barrie, 1990). We detected high concentrations (199 – 786 µg L$^{-1}$) of Ca$^{2+}$ on Devon Ice Cap, and calculated that the majority (99.5%) of Ca$^{2+}$ contributed to the nss component. Substantial concentrations (<2.00 – 78.3 µg L$^{-1}$) of other mineral dust (Al$^{3+}$, Fe$^{3+}$, Si$^{4+}$) tracers were also detected. Correlations between nss-Ca$^{2+}$ and PFAAs suggest there is a relationship between the transport of mineral dust and PFAAs to the Devon Ice Cap.



This could be caused by both mineral dust and PFAAs originating from the same regions. Alternatively, it could indicate a mechanistic relationship. Atmospheric acids are known to interact with mineral dust through reactive uptake (Sullivan et al., 2007). It is possible that PFAAs, as strong atmospheric acids, could behave in the same way and be taken up onto mineral dust aerosols and subsequently transported to the Arctic. We cannot distinguish between these mechanisms at this time, and

suggest that further studies explore this relationship between mineral dust and PFAA LRT.

## 4 Conclusions

Monitoring of temporal trends in persistent organic pollutant deposition in remote areas is an important initiative to determine the impact of regulation on contamination of pristine environments. This study demonstrates the value of ice cores to understanding contaminant LRT. Herein we report the first multi-decadal record of long-chain PFAA deposition in the

Canadian Arctic. Continuous and increasing deposition of many PFAAs on Devon Ice Cap was observed, suggesting on-going emission and use of these compounds, their precursors and likely new unidentified compounds. These results indicate that Devon Ice Cap is likely impacted by global pollutants from both North American and Eurasian Sources, with Continental Asia becoming a greater contributor in recent years. Use of major ion tracers provided new information regarding the transport of PFAAs, confirming that marine aerosol inputs are unimportant and suggesting a relationship with

mineral dust. We also observed that a small percentage of fluoride detected in the Arctic could be coming from the reactive mechanisms forming the PFAAs. Further efforts are necessary to continue monitoring the long-range transport of PFAAs and their deposition to the remote Canadian Arctic. It is important to understand these LRT mechanisms and determine the geographical sources of PFAAs and their precursors. Ice cores can aid in elucidating these mechanisms and further collection and analysis of ice cores is recommended.

**Supplement Link**

**Author Contribution**

H.M.P sectioned, extracted and analyzed the ice core, conducted the data analysis and wrote the manuscript. C.J.Y. designed and led the project, directed ice core sectioning, assisted with data analysis, and edited the manuscript. A.D.S. assisted in ice core analysis, data interpretation and edited the manuscript. C.S. aided in ice core extraction and conducted analysis. A.S.C.

conducted transport modelling and A.S.C. and M.J.S. coordinated field operations, collected the ice core, and performed the dating of the ice core. D.C.G.M. provided input on the manuscript.





**Competing Interests**

The authors declare that they have no conflict of interest.

**Acknowledgements**

This work was funded by the Northern Contaminants Program (Indigenous and Northern Affairs Canada) to C.J.Y, A.D.S,

D.C.G.M, and M.J.S., and Natural Resources Canada (Polar Continental Shelf Project) to A.S.C. We also acknowledge NSERC Discovery Grants to C.J.Y. and M.J.S. and Northern Research Supplement to M.J.S. We thank Colleen Mortimer and Anja Rutishauser for assistance with ice core collection; John MacInnis and Cyril Cook for assistance with ice core sectioning; and Trevor Vandenboer and Jamie Warren for assistance with ion chromatography analysis.

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
