# Peer review of "Continuous Non-Marine Inputs of Per- and Polyfluoroalkyl Substances to the High Arctic: A Multi-Decadal Temporal Record"

_Atmospheric Chemistry and Physics, 2017_

## Referee Comment (RC1) · J. Franklin (Referee) · 14 Dec 2017

General comments

This manuscript reports on a study of the levels of perfluoroalkyl acids (PFAAs) in a Canadian Arctic ice cap. The overall aim is to establish a temporal record of the annual deposition fluxes of PFAAs in this area, roughly locate the source regions of the incoming air masses, and determine whether the PFAAs have been transported over long distances "as such" or formed in the atmosphere by transformation of precursor compounds.

[Figure]

Although two previous studies on these questions have been performed and reported by teams including scientists belonging to the present group of authors, this latest study represents considerable progress since (a) it covers a time-span of 38 years of deposition, compared to 10-14 years for the earlier observations; (b) the sampling approach has been improved, by drilling an ice core, rather than by horizontal collection from the sidewall of a snow pit; (c) the analytical methods were upgraded, both in terms of instrument detection limits and accuracy; (d) analysis of major anions and cations in the ice confirmed that marine aerosol inputs, hypothesized by others, are unimportant to the long-range transport mechanisms of the compounds studied.

On the basis of these procedural improvements and new conclusions, I recommend publication of this multi-decadal study, after suggested minor changes as detailed below.

Specific comments on the main text

P1, L10: Replace "bioaccumulative" by "in some cases bioaccumulative".

P1, L12: After "transport" insert "(LRT)" since this acronym is used later in the text and should therefore be defined here, after the first occurrence of "long-range transport".

P1, L13: After "PFAAs" insert "in polar or mountainous regions" or some such phrase, since their usefulness is not globally ubiquitous.

P1, L18: Typo – "perfluorooctane"

P1, L19: I feel that it is important to bear in mind here that the regulations and agreements were not only North American in their extent and that deposition on the Devon Ice Cap would not be expected to be affected solely by emissions from North America, as explained later in the manuscript. The 2006 PFOA Stewardship Program, concluded by US EPA with eight leading companies in the PFAS industry, targeted the global activities of these companies pertaining to PFOA, its precursors and their longer-chain homologues. Furthermore: (a) PFOS was added in 2009 to the global Stockholm

Convention on Persistent Organic Pollutants; (b) PFOA is now restricted under the European Union's REACH Regulation as a Substance of Very High Concern; (c) PFOS is a Priority Substance under the E.U. Water Framework Directive; etc.

P1, L20: I would replace "emission and use" by "manufacture, use and emission".

P2, L3: After "(PFAAs)" insert ", a sub-group of PFASs,"

P2, L9: It would be preferable to say "This can occur through the atmosphere, in the gaseous or particle phase (including with marine aerosols), or via oceanic water currents". I believe that the marine aerosol hypothesis was first postulated in the peer-reviewed literature in the paper by Prevedouros et al (2006) cited in this manuscript, although – to my knowledge – it first originated in Poster ENV016 by Robert L. Waterland et al. at the 2005 FLUOROS Symposium in Toronto, so perhaps some credit is due here.

P2, L11: Replace "These compounds" by "These PFAAs", to avoid any confusion with the last-mentioned nouns, i.e. the PFAA precursors.

P2, L14: It would be preferable to use acronyms inspired by those recommended by Buck et al. (2011), namely (N-alkyl) FASAs and FASEs, since (for example) elsewhere in this manuscript perfluorooctane sulfonamide is abbreviated to FOSA (P4, L27 and other occurrences).

P2, L14: To avoid any confusion with fluids used for transferring heat without any change of phase (which never constituted an important market sector for CFCs), I would refer here to "fluids used for refrigeration or air conditioning" . . . if it is necessary to refer to them at all. Indeed, some CFC replacements (hydrochlorofluorocarbons and hydrofluorocarbons) used in such applications (as well as for polymeric foam blowing or as solvents) have been shown by laboratory studies to degrade in the atmosphere to give trifluoroacetic acid or, more rarely, pentafluoropropionic acid, but these shortest-chain PFCAs, although present in the ice samples, are not discussed in the present

manuscript anyway. So it is probably sufficient to restrict the discussion on precursors to FTOHs, FASAs and FASEs.

P2, L20: See comment above on marine aerosols.

P3, L9-10: The 15.5-meter ice core corresponded to 38 years of deposition, representing on average about 0.4 m per year. But the core was first separated into 1-meter sections, representing therefore some 2.5 years each. It would be useful to give a brief description of how these lengths were sectioned into discrete samples corresponding to individual years, without loss of critical fragments.

P4, L27 and 29: To avoid any confusion, the use of the terms "long-chain" and "short-chain" when referring to PFAAs here and elsewhere (e.g. Supplementary Information, P10) should comply with OECD recommendations. See the Buck et al (2011) citation, page 517.

P6, L23: Typo – Longyearbyen (also in Table S12).

P6, L24: What is meant here by "short-chain PFCAs"?

P7, L14: What was "unusual"? The PFAA concentrations, fluxes, or homologue profiles?

P8, L3: Replace "more than" by "at least".

P8, L13-14: Although Plassmann et al. (2011) did indeed state that "During melt, the timing of a chemical's release from the snowpack depends on its water solubility and its sorption to the surface of snow crystals or particles present in the bulk snow", for the PFAAs they discussed elution during melting more in terms of chain length and affinity for snow-grain surfaces or particles than of solubility.

P9, L7-8: The references cited refer to animal biota. It would be useful to add one or more references for human blood, e.g. from the NHANES studies (Calafat and co-workers).

P9, Figure 2: It does not seem reasonable to label the right-hand ordinate axes "emissions". The figures refer to production tonnages of polymer products, which may never be exposed to the environment in such a manner that any non-polymeric trace components may leach out or exude.

P9, Figure 2 and ensuing paragraph: However, just comparing ice-cap deposition fluxes of PFOA or PFNA to production volumes of various fluoropolymers makes no sense when one takes into account the following facts: (a) much PTFE and PFNA (and no doubt also FEP and PFA) is manufactured by the suspension polymerization process, which does not require the use of fluorinated surfactants as processing aids, unlike the emulsion (or dispersion) process; (b) PFOA and PFNA were phased out and replaced by alternative processing aids by all the major manufacturers in the developed countries long before 2015, in order to meet the objectives of US EPA's 2006 PFOA Stewardship Program. Useful quantitative information on production volumes for different fluoropolymer types could no doubt be found, if required, in IHS Markit's 2016 Chemical Economics Handbook report on Fluoropolymers (https://www.ihs.com/products/fluoropolymers-chemical-economics-handbook.html); (c) fluoropolymer manufacturers have taken steps to drastically reduce the levels of fluorinated processing aids in their final products, for those fluoropolymers made by the dispersion process; (d) as discussed in this manuscript, environmental emissions of PFOA and PFNA are not related solely to the manufacture and use of (certain) fluoropolymers, but also arise from the degradation of precursors, such as FTOHs and derivatives thereof. I would therefore strongly recommend deleting the misleading fluoropolymer production curves from Figure 2.

P10, L1 and 2: Use "PFASs" as the plural of "PFAS".

P10, L18-19: The anomalous value of 80 ng m-2 yr-1 is indeed plotted for the year 2012 in Figure 3, but it is listed in Table S11 for 2013.

P10, L22-23: "This suggests FOSA is not the primary source of PFOS to the Devon Ice

[Figure]

Cap". I do not feel that the lack of correlation is very significant. If there are several distinct spatial pathways for transport of air masses from sources regions to the Devon Ice Cap, the degree of atmospheric transformation of FOSA along the trajectory (for which the lifetime is not discussed in this manuscript) may vary with the individual pathway, so that the FOSA/PFOS ratio on arrival may depend not only on the corresponding ratio in the source region, but also on the pathway followed. In any case, it is reasonable to assume that there are sources of PFOS ("as such") that are quite independent of FOSA or other sulfonamido derivatives.

P11, L8-9: If POSF-based products were replaced by PBSF-based ones after 2002, is it not surprising that PFBS was detected in the ice samples only in two recent years? Does this not warrant a comment?

P12, L8: "The decline of the production of the precursor FOSA by 3M". Should this be interpreted as meaning the "decline of precursor POSF production, and hence of all C8 sulfonamido derivatives, including FOSA"?

P12, L28, 29, 30 and 31: "2013" or "2012"?

P13, L6: Change acronyms for NAFSEs and NAFSEs.

P13, L17: Typo – Longyearbyen

P13, L31: Replace "compound" by "homologue".

P14, L13: I do not feel that it has been demonstrated that direct transport of PFOS and other PFSAs to the ice cap is unlikely.

P14, L14: What is meant by "and to the atmosphere into the ocean", since deposition from the atmosphere to any oceanic system has already been mentioned in the same sentence?

P14, L14-15: "and then re-enter the atmosphere from oceanic currents and marine aerosols". Are the fully ionized PFAAs likely to partition from the ocean to the atmospheric compartment other than in marine aerosol particles?

P14, L16-17: Insert "PFAAs" between "and" and "may" (i.e. when on aerosols, they are no longer "gas-phase").

P15, L11: Typo – Arctic

P15, L13 and 14: Change "concentrations" to "fractions" or "proportions"?

P15, L15: I do not understand the claim that "indirect sources are of importance for Arctic deposition" since PFHxS was not found on the ice cap. Does "indirect sources" refer to ocean transport?

P15, Figure 5: Are the Devon Ice Cap values taken solely from the present study, or are previous measurements included (for 2005, at least)?

P16, L8: See previous comment on "heat transfer fluids".

P17, L10-11: This sentence needs some rephrasing. What seems obvious is that there are ongoing emissions of many PFAAs and/or their precursors and that they arrive through the atmosphere as far as the Devon Ice Cap. The "use" of PFAAs is not so evident. Long-chain ones and their precursors are probably still used in the less-developed countries, while the short-chain homologues have largely replaced them in the developed world. What is intended by "likely new unidentified compounds" if they are not included in "PFAA precursors"?

P17, L16: After "PFAAs" insert "in the atmosphere".

P20, L16: Perhaps "ECCC" should be spelled out for those readers not familiar with the acronym. Also, it would be useful to provide a web-link, as in the Supplementary Information.

P21, L28: Typo – Kallenborn

P24, L5-6: This paper has now been published in Atmospheric Chemistry and Physics,

so the citation should be replaced by that of the final published version.

P24, L7: It would be useful to provide a web-link, as in the Supplementary Information.

Specific comments on the Supplementary Information

P8, Table S4: The concentration units should be indicated.

P12, first line of text: Since FOSA is neither a PFCA nor a PFSA, "including" should be replaced by "as well as" or "together with".

P17, penultimate line: "albeit with some subtle differences".

P19: On this right-hand page, the figure does not extend until the year 2015 and data for some PFCAs (e.g. PFNA) appear to be missing even for 2014. Similarly, on the left-hand panel (P18) some data (PFOA, PFHxA) listed in Table S10 for 2015 are not plotted on the chart.

P21, Table S12: Would not this table be more readable if the concentrations were expressed in pg/L, as in Tables S8 and S9?

P23, Table S13 and P24, Table S14: Does N/A mean "not available" or "not applicable"? In Table S14, does "non-US" include Canada?

P25, Table S15: The title should explicitly include FOSA, which is not a PFAA homologue.

P33, 2nd Kwok reference: Typo – Kallenborn
* * *

---

## Referee Comment (RC2) · Anonymous Referee #2 · 24 Jan 2018

This is an interesting effort to measure and discuss the PFAS concentrations and fluxes at a multidecadal scale from an ice core in the high arctic. The topic is of great importance, and the data set may be unique. I disagree with an important fraction of the discussion (see bellow). Briefly, important issues are, i) a better assessment of discussion of blanks, ii) a complete description of the data set, iii) a different discussion of air mass back trajectories, iv) a re-evaluation of the processes responsible to transfer PFAS from the air to snow, including the post depositional processes and the assessment of the role of sea-salt aerosol. Therefore, I think that this manuscript needs major revision before it can be accepted.

[Figure]

- Page 4, line 5. The assessment of blanks is weak. Blanks were performed for the cartridges, but this only covers the potential contamination during analysis in the laboratory. There is no assessment of field blanks, as samples could be contaminated during sampling, handling, transport, and sectioning. This needs some comments.

- Page 5, line 13. As I understand the back trajectories provided by Hysplit, there is an increasing uncertainty for longer time periods. While the 48 hours back trajectories are reliable, there is a huge uncertainty for 10 day back trajectories. I suggest to shorten the back trajectories and discuss the uncertainties.

- Page 6, line 5. Report and discuss the method for dating the subsamples from the ice core and determining the annual snow accumulation.

- Page 6, line 15. "The most…." This sentence needs a reference or justification.

- Page 6, line 21…. Compare as well with concentrations and fluxes from maritime Antarctica (Casal et al. EST 2017).

- Section 3.1 I suggest to include one or two figures with the patterns, concentrations or fluxes, and extend the discussion. There are some differences in the patterns reported for snow (fresh and/or accumulated) and it is not clear how the results shown here fit with previous results. For me, the major contribution of this manuscript is the data set of the measurements, much more important than the modelling, however, the paper does not reflect this.

- Page 7, line 10. Provide information for the other years.

- Figure 1 and page 7. I don't agree with the discussion and conclusions for the results shown in figure 1. There is a very weak evidence for source regions in part of Asia, first, due to the uncertainty of the analysis, and secondly, because the signaled regions have a very low population. Figure 1 shows that the Arctic ocean (ice covered or not) may be the main source regions. I wonder if specific ice-influenced photochemistry may affect the formation of the targeted compounds.

- I would suggest to try to correlate the concentrations and fluxes with the extension of the arctic ice cap.

- Page 13, lines 4-5. I don't understand this sentence.

- An implicit assumption in this work is that the observed pattern and concentrations are a direct consequence of atmospheric snow deposition, thus snow scavenging of atmospheric PFAS. No discussion is made of pot-depositional processes affecting the concentrations and patterns of PFAS. - Even If I was convinced that most PFAS in the studied arctic region come from atmospheric

oxidation of neutral precursors, no discussion is made on the processes responsible for the transport of ionizable PFAS from the gas phase (I guess oxidation occurs in the gas phase) to deposited snow. Once a ionizable PFAS is formed, it may attach rapidly to aerosols. Which are the dominant aerosol types in this region? A reference is needed to support the response to this question. Furthermore, it could be that neutral precursors are the main contributors to ionizable PFAS in the surface Arctic ocean, and then these are transferred to Devon Cap by sea-salt aerosol formation and deposition.

- Page 15, line 5. I don't know any study on the occurrence of PFAS in the marine surface microlayer, but for other POPs, there is a huge variability on the enrichment factors, thus a lack of correlation does not contradict the potential role of marine aerosols. Furthermore, Na can be forced to move out from the snow/ice after deposition.

- Page 15, line 9. This is not true because the patterns in snow are different than in seawater even for a given site, and because we do not know the patterns in the surface microlayer, nor in remote aerosols.

- I suggest to plot the ratio of concentrations between Devon cap and arctic ocean (excep for PFBA which has clearly a different origin and behavior), and plot this ratio versus the number of C of the pfas chain, and then discuss taking into account the literature.
- The authors insist that sea salt aerosol does not play a role, but I don't see concluding arguments. Which is the main source of aerosols in the region and how ionizable pfas behave after their formation? Ok, let's assume that sea-salt aerosols do not play an important role, then, to which aerosols are PFAS bound to? Are they in the gas phase and then scavenged by snow? Please, provide a plausible mechanistic explanation.

- Pages 16-17. A hypothetical dust source is commented, but this is not supported by the assessment of air mass back trajectories.

- Note that Ca and Mg are enriched in sea-salt aerosol coming from the sea surface microlayer (Jayarathne et al. EST 2016)! Then the correlations between Ca and PFAS would support sea salt as an important contributor to PFAS at the studied site!

- After reading this manuscript I think that the interpretation needs to be re-evaluated, and a new version prepared taking into account my comments above.

---

## Author Comment (AC1) · 7 Mar 2018

We thank Dr. Franklin for his constructive comments on our manuscript. Please find our responses below in yellow, with changes to the manuscript indicated in **bold**.

General comments
This manuscript reports on a study of the levels of perfluoroalkyl acids (PFAAs) in a Canadian Arctic ice cap. The overall aim is to establish a temporal record of the annual deposition fluxes of PFAAs in this area, roughly locate the source regions of the incoming air masses, and determine whether the PFAAs have been transported over long distances "as such" or formed in the atmosphere by transformation of precursor compounds.

Although two previous studies on these questions have been performed and reported by teams including scientists belonging to the present group of authors, this latest study represents considerable progress since (a) it covers a time-span of 38 years of deposition, compared to 10-14 years for the earlier observations; (b) the sampling approach has been improved, by drilling an ice core, rather than by horizontal collection from the sidewall of a snow pit; (c) the analytical methods were upgraded, both in terms of instrument detection limits and accuracy; (d) analysis of major anions and cations in the ice confirmed that marine aerosol inputs, hypothesized by others, are unimportant to the long-range transport mechanisms of the compounds studied. On the basis of these procedural improvements and new conclusions, I recommend publication of this multi-decadal study, after suggested minor changes as detailed below.

Specific comments on the main text
1) P1, L10: Replace "bioaccumulative" by "in some cases bioaccumulative".
Response: We have added "in some cases" before the word "bioaccumulative". Now it reads: "Perfluoroalkyl acids (PFAAs) are persistent, **in some cases** bioaccumulative compounds found ubiquitously within the environment." (P1, L10-L11)

2) P1, L12: After "transport" insert "(LRT)" since this acronym is used later in the text and should therefore be defined here, after the first occurrence of "long-range transport".
Response: We have added the acronym "(LRT)" after the first occurrence of the word "long-range transport". (P1, L12)

3) P1, L13: After "PFAAs" insert "in polar or mountainous regions" or some such phrase, since their usefulness is not globally ubiquitous.
Response: We have added "in polar or mountainous regions" after the word "PFAAs". Now it reads:
"Ice caps preserve a temporal record of PFAA deposition making them useful in studying the atmospheric trends in LRT of PFAAs **in polar or mountainous regions**, as well as understanding major pollutant sources and production changes over time." (P1, L12-L14)

4) P1, L18: Typo – "perfluorooctane"
Response: We have fixed the typo and changed the word from "perflurooctane" to "perfluorooctane". (P1, L19)

5) P1, L19: I feel that it is important to bear in mind here that the regulations and agreements were not only North American in their extent and that deposition on the

Devon Ice Cap would not be expected to be affected solely by emissions from North America, as explained later in the manuscript. The 2006 PFOA Stewardship Program, concluded by US EPA with eight leading companies in the PFAS industry, targeted the global activities of these companies pertaining to PFOA, its precursors and their longer-chain homologues. Furthermore: (a) PFOS was added in 2009 to the global Stockholm Convention on Persistent Organic Pollutants; (b) PFOA is now restricted under the European Union's REACH Regulation as a Substance of Very High Concern; (c) PFOS is a Priority Substance under the E.U. Water Framework Directive; etc.

Response: We thank Dr. Franklin for making this point. It is true that we explain this later in the manuscript, but we should make it clearer in this sentence in the abstract. We have changed the sentence to include the word "international" and not just "North American". Now it reads:
"Our results demonstrate that the PFCAs and perfluorooctane sulfonate (PFOS) have continuous and increasing deposition on Devon Ice Cap, despite recent North American **and international** regulations and phase-outs." (P1, L18-L20)

6) P1, L20: I would replace "emission and use" by "manufacture, use and emission".
Response: We have replaced "emission and use" with "manufacture, use and emissions". Now it reads:
"We propose that this is the result of on-going **manufacture, use and emissions** of these compounds, their precursors and other newly unidentified compounds in regions outside of North America." (P1, L20-L22)

7) P2, L3: After "(PFAAs)" insert ", a sub-group of PFASs,"
Response: We have inserted ", a sub-group of PFASs" after "(PFAAs)". Now it reads:
"Perfluoroalkyl acids (PFAAs), **a sub-group of PFASs,** are persistent contaminants that are ubiquitous in the environment." (P2, L3-L4)

8) P2, L9: It would be preferable to say "This can occur through the atmosphere, in the gaseous or particle phase (including with marine aerosols), or via oceanic water currents". I believe that the marine aerosol hypothesis was first postulated in the peer reviewed literature in the paper by Prevedouros et al (2006) cited in this manuscript, although – to my knowledge – it first originated in Poster ENV016 by Robert L. Waterland et al. at the 2005 FLUOROS Symposium in Toronto, so perhaps some credit is due here.
Response: We thank Dr. Franklin for his suggestion for reformatting this sentence and who to properly cite for this fact. We have now cited Prevedouros et al (2006) and the sentence now reads:
"This can occur **through the atmosphere, in the gaseous or particle phase (including with marine aerosols), or via oceanic water currents (Prevedouros et al., 2006)**. " (P2, L9-L10)

9) P2, L11: Replace "These compounds" by "These PFAAs", to avoid any confusion with the last-mentioned nouns, i.e. the PFAA precursors.
Response: We have changed the word "compounds" to the word "PFAAs". Now it reads:
"These **PFAAs** are environmentally persistent and longer chain acids (>6 carbons) have a tendency to bioaccumulate and biomagnify in food webs." (P2, L12-L13)

10) P2, L14: It would be preferable to use acronyms inspired by those recommended by Buck et al. (2011), namely (N-alkyl) FASAs and FASEs, since (for example) elsewhere in this manuscript perfluorooctane sulfonamide is abbreviated to FOSA (P4, L27 and other occurrences).

Response: We have changed the acronyms from "NAFSAs/NAFSEs" to (**N-Alkyl) FASAs/FASEs**, to avoid any confusion with acronyms throughout the paper.

11) P2, L14: To avoid any confusion with fluids used for transferring heat without any change of phase (which never constituted an important market sector for CFCs), I would refer here to "fluids used for refrigeration or air conditioning" : : : if it is necessary to refer to them at all. Indeed, some CFC replacements (hydrochlorofluorocarbons and hydrofluorocarbons) used in such applications (as well as for polymeric foam blowing or as solvents) have been shown by laboratory studies to degrade in the atmosphere to give trifluoroacetic acid or, more rarely, pentafluoropropionic acid, but these shortest chain PFCAs, although present in the ice samples, are not discussed in the present manuscript anyway. So it is probably sufficient to restrict the discussion on precursors to FTOHs, FASAs and FASEs.

Response: We thank Dr. Franklin for this great comment. Since these CFC-replacements will be discussed in a separate paper, we have decided to remove any reference to the "CFC-replacements" or "heat-transfer fluids" from this discussion. This sentence now reads:
"In the atmosphere, volatile and semi-volatile precursors such as fluorotelomer alcohols (FTOHs) and N-alkyl perfluoroalkane sulfonamides/sulfonamidoethanols ((N-Alkyl) FASAs/FASEs) undergo oxidation in the gas phase to form PFAAs (D'eon et al., 2006; Ellis et al., 2004; Young and Mabury, 2010)." (P2, L14-L16)

12) P2, L20: See comment above on marine aerosols.

Response: Since the main focus of this paper is on atmospheric inputs of PFAAs and not oceanic inputs, we have decided to remove the short paragraph on marine aerosols, for better clarity. The paragraph and associated references we removed from the introduction reads as follows: "Further transport can occur via ocean currents (Armitage et al., 2006, 2009a, 2009b) and marine aerosols (McMurdo et al., 2008). PFAAs are highly acidic, surface-active compounds, usually present as anions in the aqueous phase under environmental conditions (Cheng et al., 2009). These surface-active compounds will concentrate at the air-water interface and are therefore expected to be in the sea surface microlayer (SSML) and to be present in marine aerosols (Lewis and Schwartz, 2004)." (P2, L20)

13) P3, L9-10: The 15.5-meter ice core corresponded to 38 years of deposition, representing on average about 0.4 m per year. But the core was first separated into 1-meter sections, representing therefore some 2.5 years each. It would be useful to give a brief description of how these lengths were sectioned into discrete samples corresponding to individual years, without loss of critical fragments.

Response: We thank Dr. Franklin for bringing this up and we have added a paragraph into Section S1 of the SI, describing how the ice core was sectioned and bottled into individual years, with minimal loss and contamination. The paragraph reads as follows:
    "**The 15.5-meter ice core was separated and packaged into 1-meter sections for transport. Extensive care was taken in handling the ice core to avoid any introduction of**

contamination that could compromise the trace analysis. During collection process, handling, and sample preparation, no products containing fluoropolymer coatings came into contact with the ice cores. Additionally, care was taken to scrape the potentially contaminated outer "rind" of the core sections, using only the inner uncontaminated ice for analysis. Depths corresponding to calendar years were determined as described above. Using this data, the dates for each 1-meter section were determined. For sectioning the 1-meter ice cores into the equivalent years, we removed each 1-meter section from the packaging, and placed the firn and ice pieces onto aluminum foil, cleaned with methanol. We cut the firn or ice pieces at the depth that corresponded to each year with a pre-cleaned saw and then added the sectioned firn and ice for one year into a labeled pre-cleaned polypropylene bottle. Using the aluminum foil underneath the firn and ice pieces, we were able to easily pick up the foil and pour the firn and ice pieces into the labeled bottle, with minimal loss." (P4-P5)

14) P4, L27 and 29: To avoid any confusion, the use of the terms "long-chain" and "shortchain" when referring to PFAAs here and elsewhere (e.g. Supplementary Information, P10) should comply with OECD recommendations. See the Buck et al (2011) citation, page 517.

Response: We thank Dr. Franklin for bringing up the use of these terms in this comment. It is difficult for our approach in this paper to use the terms "long-chain" and "short-chain" in accordance with the OECD described in Buck et al (2011), as we are defining these terms in this paper in terms of source/mechanism of formation. However, to try and avoid confusion, we have reworded sentences where we either avoid using the term "short or long" altogether, or define the PFAA chain-lengths that we are referring to for that individual sentence. In this first instance (P4, L29-L31) we replaced "long-chain PFAA analysis" and "short-chain PFAA analysis" with "**first** PFAA analysis" and "**second** PFAA analysis" and defined the chain-lengths we analyzed in brackets.

15) P6, L23: Typo – Longyearbyen (also in Table S12).
Response: We have corrected "Longyearbreen" to "**Longyearbyen**". (P6, L27 and Table S12)

16) P6, L24: What is meant here by "short-chain PFCAs"?
Response: We have removed this sentence. (P6, L28)

17) P7, L14: What was "unusual"? The PFAA concentrations, fluxes, or homologue profiles?
Response: We thank Dr. Franklin for asking about this point, as using the word "unusual" is not clear in this sentence. We are referring to the high PFOS flux that was observed in 2013. To make the sentence clearer, it now reads:
"Because of **the high PFOS flux observed** in 2013, we also examined this year to identify any transport anomalies, but found both years (1994 and 2013) to be comparable." (P7, L11-L13)

18) P8, L3: Replace "more than" by "at least".
Response: We have replaced the words "more than" with "at least". Now it reads:
"Temporal trends of shorter-chain PFCAs (C2 – C4) will be discussed separately, so this section will focus solely on PFCAs with **at least** five carbons." (P8, L2-L3)

19) P8, L13-14: Although Plassmann et al. (2011) did indeed state that "During melt, the timing of a chemical's release from the snowpack depends on its water solubility and its sorption to the surface of snow crystals or particles present in the bulk snow", for the PFAAs they discussed elution during melting more in terms of chain length and affinity for snow-grain surfaces or particles than of solubility.

Response: We thank Dr. Franklin for this comment. It is true that Plassmann et al. (2011) mention not just water solubility but also chain length in their PFAA elution discussion. We have changed this sentence to include these other factors of PFAA elution. The sentence now reads: "One study has examined elution behaviour of PFAAs from a melting snowpack and found that elution of PFAAs is driven by **a number of factors including** water solubility in the snowpack, **PFAA chain length, and sorptive capacity of the snow grain surface** (Plassmann et al., 2011)." (P8, L12-L14)

20) P9, L7-8: The references cited refer to animal biota. It would be useful to add one or more references for human blood, e.g. from the NHANES studies (Calafat and coworkers).

Response: We thank Dr. Franklin for making this good point and for offering a reference suggestion. We have added Calafat et al. (2007) and Olsen et al. (2003) as references to human serum and changed the words "human blood" to "human serum". The sentence now reads: "This is evident in temporal trend analysis in Canada and the U.S. such as in human **serum**, freshwater fish, and non-migratory birds, which do not show much of a decline in PFCAs from 1990s to 2012 (Braune and Letcher, 2013; **Calafat et al., 2007**; Gewurtz et al., 2016; **Olsen et al., 2003)**." (P9, L12-L14)

21) P9, Figure 2: It does not seem reasonable to label the right-hand ordinate axes "emissions". The figures refer to production tonnages of polymer products, which may never be exposed to the environment in such a manner that any non-polymeric trace components may leach out or exude.

Response: In response to comment 22, we have removed the production data, which means there is no right-hand axis now. (P10, Figure 2)

22) P9, Figure 2 and ensuing paragraph: However, just comparing ice-cap deposition fluxes of PFOA or PFNA to production volumes of various fluoropolymers makes no sense when one takes into account the following facts: (a) much PTFE and PFNA (and no doubt also FEP and PFA) is manufactured by the suspension polymerization process, which does not require the use of fluorinated surfactants as processing aids, unlike the emulsion (or dispersion) process; (b) PFOA and PFNA were phased out and replaced by alternative processing aids by all the major manufacturers in the developed countries long before 2015, in order to meet the objectives of US EPA's 2006 PFOA Stewardship Program. Useful quantitative information on production volumes for different fluoropolymer types could no doubt be found, if required, in IHS Markit's 2016 Chemical Economics Handbook report on Fluoropolymers (https://www.ihs.com/products/fluoropolymerschemical-economics-handbook.html); (c) fluoropolymer manufacturers have taken steps to drastically reduce the levels of fluorinated processing aids in their final products, for those fluoropolymers made by the dispersion process; (d) as discussed in this manuscript, environmental emissions of PFOA

and PFNA are not related solely to the manufacture and use of (certain) fluoropolymers, but also arise from the degradation of precursors, such as FTOHs and derivatives thereof. I would therefore strongly recommend deleting the misleading fluoropolymer production curves from Figure 2.

Response: We thank Dr. Franklin for this significant comment regarding Figure 2. We understand how plotting production volumes of fluoropolymers might not be the best representation of what we were trying to explain, and so we have removed this data from Figure 2. What we were trying to do was compare production volumes from Wang et al. (2014) (which we could openly access) to PFOA and PFNA to just get an idea of how production volume is increasing in China, but not in the other previous production countries (Europe, Japan, USA). And to add to our discussion about how the ongoing deposition of these PFAAs is likely also coming from China and not just the other countries historically known for manufacturing these fluoropolymers. But again, as Dr. Franklin notes, this fluoropolymer data is not the best representation. Ideally, it would be very beneficial to compare quantitative data on production volumes for different fluoropolymer types to our PFAAs, but information such as this is not open access, including the IHS Markit's 2016 Chemical Economics Handbook report on Fluoropolymers that Dr. Franklin recommends, which is unfortunately very expensive to purchase. To avoid misrepresentation, we have removed this information from Figure 2. (P10)

23) P10, L1 and 2: Use "PFASs" as the plural of "PFAS".

Response: We have corrected this so that when we refer to the plural of "PFAS", it is now written as "PFASs". (P10, L5-L6)

24) P10, L18-19: The anomalous value of 80 ng m-2 yr-1 is indeed plotted for the year 2012 in Figure 3, but it is listed in Table S11 for 2013.

Response: We thank Dr. Franklin for pointing this error out. That was an error on our part in plotting the data in terms of both depth and year. We have fixed this in figure 3, which now shows the high PFOS flux for the year 2013, and have made sure that we refer to 2013 throughout the paper when discussing the 80 ng m-2 yr-1 anomalies. (P11, L12) (P12, Figure 3)

25) P10, L22-23: "This suggests FOSA is not the primary source of PFOS to the Devon Ice Cap". I do not feel that the lack of correlation is very significant. If there are several distinct spatial pathways for transport of air masses from sources regions to the Devon Ice Cap, the degree of atmospheric transformation of FOSA along the trajectory (for which the lifetime is not discussed in this manuscript) may vary with the individual pathway, so that the FOSA/PFOS ratio on arrival may depend not only on the corresponding ratio in the source region, but also on the pathway followed. In any case, it is reasonable to assume that there are sources of PFOS ("as such") that are quite independent of FOSA or other sulfonamido derivatives.

Response: We thank Dr. Franklin for this insightful and important comment. We have removed the phrase "This suggests FOSA is not the primary source of PFOS to the Devon Ice Cap" and replaced it with:
"**Therefore, it is reasonable to assume that there are sources of PFOS that are independent of FOSA or other sulfonamido derivatives**." (P11, L15-L17)

26) P11, L8-9: If POSF-based products were replaced by PBSF-based ones after 2002, is it not surprising that PFBS was detected in the ice samples only in two recent years? Does this not warrant a comment?

Response: We thank Dr. Franklin for making an excellent point here. We have added a comment about PFBS detection in this paragraph, as well as a short statement about how our understanding of the LRT mechanisms of these PFASs is incomplete and needs further study. This sentence reads:

**"This is interesting because we actually detected quantifiable levels of PFBS in the ice core in two recent years, and so our understanding of LRT of these PFASs is incomplete and warrants further research."** (P13, L1-L3)

27) P12, L8: "The decline of the production of the precursor FOSA by 3M". Should this be interpreted as meaning the "decline of precursor POSF production, and hence of all C8 sulfonamido derivatives, including FOSA"?

Response: We thank Dr. Franklin for suggesting clearer phrasing. The sentence has been reworded:

**"The decline of precursor POSF production, and hence of all C8 sulfonamido derivatives, including FOSA, by 3M**…" (P13, L7-L8)

28) P12, L28, 29, 30 and 31: "2013" or "2012"?

Response: We thank Dr. Franklin for pointing out this error. The high PFOS flux was in 2013, which is correctly specified in this paragraph. We have fixed this error elsewhere, in Figure 3 and P11, L12 where it stated and showed 2012.

29) P13, L6: Change acronyms for NAFSEs and NAFSEs.

Response: We have changed the acronyms from "NAFSAs and NAFSEs", to "**(N-Alkyl) FASAs/FASEs**". (P14, L6)

30) P13, L17: Typo – Longyearbyen

Response: We have corrected "Longyearbreen glacier" to "**Longyearbyen, Svalbard**". (P14, L17)

31) P13, L31: Replace "compound" by "homologue".

Response: We disagree with Dr. Franklin here. In this paragraph we are referring to the pairs of PFCAs as "homologues" and the n:2 fluorotelomers as "compounds". Therefore, on P14, L31, we are referring to the 8:2 FTOH "compound", not "homologue".

32) P14, L13: I do not feel that it has been demonstrated that direct transport of PFOS and other PFSAs to the ice cap is unlikely.

Response: We thank Dr. Franklin for making this important point. We have changed the language of this sentence to indicate that direct transport may act as a minor source of PFAAs to the ice cap. The sentence now reads:

"Thus far, evidence **supports** indirect formation of PFAAs in the atmosphere from volatile precursor compounds **as the major source**, indicating that direct transport is **likely a minor source** of PFAAs to the ice cap." (P15, L12-L13)

33) P14, L14: What is meant by "and to the atmosphere into the ocean", since deposition from the atmosphere to any oceanic system has already been mentioned in the same sentence?

Response: We thank Dr. Franklin for bringing this up. We have removed "and to the atmosphere into the ocean" from the sentence as it is a repetitive. Now it reads: "Atmospherically formed PFAAs can deposit to any terrestrial or oceanic system. **Those PFAAs deposited to oceans can** re-enter the atmosphere **in the form of** marine aerosols (McMurdo et al., 2008)." (P15, L13-L15)

34) P14, L14-15: "and then re-enter the atmosphere from oceanic currents and marine aerosols". Are the fully ionized PFAAs likely to partition from the ocean to the atmospheric compartment other than in marine aerosol particles?

Response: We thank Dr. Franklin for noting this error. We have rephrased the sentence as described above in the response to comment 33 to indicate that only marine aerosols are re-entering the atmosphere and have included the reference McMurdo et al. (2008). The sentence now reads:
"Atmospherically formed PFAAs can deposit to any terrestrial or oceanic system. **Those PFAAs deposited to oceans can** re-enter the atmosphere **in the form of** marine aerosols **(McMurdo et al., 2008)**." (P15, L13-L15)

35) P14, L16-17: Insert "PFAAs" between "and" and "may" (i.e. when on aerosols, they are no longer "gas-phase").

Response: We have added the word "PFAAs" between "and" and "may" for further clarity. The sentence now reads: "Gas-phase PFAAs will deposit through wet or dry deposition and **PFAAs may also** be transported on aerosols (Thackray and Selin, 2016)." (P15, L17)

36) P15, L11: Typo – Arctic

Response: We have corrected "Artic" to "**Arctic**". (P16, L13)

37) P15, L13 and 14: Change "concentrations" to "fractions" or "proportions"?

Response: We have changed "concentrations" to either "concentration fractions" or just "fractions". The sentence now reads:
"Molar concentration **fractions** of PFAAs on the ice cap differed from ocean PFAA **fractions**, with higher molar concentration **fractions** of PFBA and PFNA, and lower molar concentration **fractions** of PFSAs found on Devon Ice Cap." (P16, L15-L17)

38) P15, L15: I do not understand the claim that "indirect sources are of importance for Arctic deposition" since PFHxS was not found on the ice cap. Does "indirect sources" refer to ocean transport?

Response: We thank Dr. Franklin for this comment. We have clarified these couple sentences to explain why PFHxS is an important marker, and how it adds to our discussion that marine aerosols are not a major source of PFAA deposition to the Arctic environment. We also added MacInnis et al., (2017) as a reference, as they provide a more detailed explanation on the significance of PFHxS. These sentences now read:
"**As described by MacInnis et al. (2017), PFHxS can act as a marker for direct transport. The absence of PFHxS on the ice cap is further confirmation that marine aerosols are not a significant source for PFAA deposition to the Arctic environment.**" (P16, L18-L20)

39) P15, Figure 5: Are the Devon Ice Cap values taken solely from the present study, or are previous measurements included (for 2005, at least)?

Response: The Devon Ice Cap values in Figure 5 are from the present study. We have now indicated this in the description for Figure 5, to avoid confusion. The description now reads: "Molar concentration fraction of sum of PFAAs on the Devon Ice Cap **(from the current 2015 ice core collection)** compared to levels in the Canadian Arctic Archipelago (Benskin et al., 2012), Arctic Ocean (Benskin et al., 2012; Cai et al., 2012b), North Atlantic Ocean (Zhao et al., 2012), and North Pacific Ocean (Cai et al., 2012b) in 2005 and 2010." (P17, L2)

40) P16, L8: See previous comment on "heat transfer fluids".

Response: We have removed the words "heat transfer fluids". The sentence now reads: "The degradation of many PFCA precursors forms both PFCA and HF." (P17, L10)

41) P17, L10-11: This sentence needs some rephrasing. What seems obvious is that there are ongoing emissions of many PFAAs and/or their precursors and that they arrive through the atmosphere as far as the Devon Ice Cap. The "use" of PFAAs is not so evident. Long-chain ones and their precursors are probably still used in the less-developed countries, while the short-chain homologues have largely replaced them in the developed world. What is intended by "likely new unidentified compounds" if they are not included in "PFAA precursors"?

Response: We thank Dr. Franklin for this important comment. We understand how this sentence can be confusing. We've rephrased this sentence to read:
"Continuous and increasing deposition of many PFAAs on Devon Ice Cap was observed, suggesting on-going emission and use of these **PFAAs and their precursors in areas such as North America and Eurasia.**" (P18, L27-L28)

42) P17, L16: After "PFAAs" insert "in the atmosphere".

Response: We added "in the atmosphere" after "PFAAs". The sentence now reads:
"Use of major ion tracers provided new information regarding the transport of PFAAs **in the atmosphere**, confirming that marine aerosol inputs are unimportant and suggest a relationship with mineral dust." (P18, L30-L32)

43) P20, L16: Perhaps "ECCC" should be spelled out for those readers not familiar with the acronym. Also, it would be useful to provide a web-link, as in the Supplementary Information.

Response: We have changed "ECCC" to "Environment and Climate Change Canada" in the reference and provided a web-link. We also corrected this in the SI. (P22, L2-L3)

44) P21, L28: Typo – Kallenborn

Response: We have corrected "Kallerborn" to "Kallenborn". (P23, L8)

45) P24, L5-6: This paper has now been published in Atmospheric Chemistry and Physics, so the citation should be replaced by that of the final published version.

Response: We have updated this citation: "Thackray, C. P. and Selin, N. E.: Uncertainty and variability in atmospheric formation of PFCAs from fluorotelomer precursors, Atmos. Chem.

Phys., 17(7), 4585–4597, doi:10.5194/acp-17-4585-2017, 2017." (P25, L20-L21)

46) P24, L7: It would be useful to provide a web-link, as in the Supplementary Information.
Response: This citation has been updated with a web-link. (P25, L23-L24)

Specific comments on the Supplementary Information:
47) P8, Table S4: The concentration units should be indicated.
Response: We have added the concentration units of ng/L to Table S5. (P9)

48) P12, first line of text: Since FOSA is neither a PFCA nor a PFSA, "including" should be replaced by "as well as" or "together with".
Response: We have replaced the word "including", with "as well as". Now it reads:
"A number of PFCAs and PFSAs, **as well as** FOSA were detected on the Devon Ice Cap."
(P13)

49) P17, penultimate line: "albeit with some subtle differences".
Response: We have corrected the sentence. It now reads:
"The data sets generally agree albeit **with** some subtle differences (i.e. the mean +/- uncertainty overlap between the three studies), the exception being PFOS." (P18)

50) P19: On this right-hand page, the figure does not extend until the year 2015 and data for some PFCAs (e.g. PFNA) appear to be missing even for 2014. Similarly, on the left-hand panel (P18) some data (PFOA, PFHxA) listed in Table S10 for 2015 are not plotted on the chart.
Response: We have corrected these two plots in figure S2. (P19-P20) Both plots now include the year 2015 and the correct data presented in Table S11.

51) P21, Table S12: Would not this table be more readable if the concentrations were expressed in pg/L, as in Tables S8 and S9?
Response: The Reviewer makes a good point. We have changed the concentrations in Table S13 to pg/L instead of ng/L. (P22)

52) P23, Table S13 and P24, Table S14: Does N/A mean "not available" or "not applicable"?
Response: The N/A means "not available". We have added this to the footnotes in Tables S14 and S15. (P24-P25)

53) In Table S14, does "non-US" include Canada?
Response: Non-US operations in Table S15 refer to the companies located in Europe and Asia. Canada is included in Table S14. We have added "Non-U.S. and Non-Canadian operations" to the heading for Table S15 to make it clearer. (P25)

54) P25, Table S15: The title should explicitly include FOSA, which is not a PFAA homologue.
Response: We have included "and FOSA" to the title of Table S16. (P26) We have also added "as well as FOSA" to Tables S20 and S21. (P33-P34)

55) P33, 2nd Kwok reference: Typo – Kallenborn
Response: We have corrected the typo "Kallerborn" to "Kallenborn". (P37)

**References**

[revised manuscript text omitted]

---

## Author Comment (AC2) · 7 Mar 2018

We thank the reviewer for their thoughtful comments on our manuscript. Please find our responses below in yellow, with changes to the manuscript indicated in **bold**.

This is an interesting effort to measure and discuss the PFAS concentrations and fluxes at a multidecadal scale from an ice core in the high arctic. The topic is of great importance, and the data set may be unique. I disagree with an important fraction of the discussion (see below). Briefly, important issues are, i) a better assessment of discussion of blanks, ii) a complete description of the data set, iii) a different discussion of air mass back trajectories, iv) a re-evaluation of the processes responsible to transfer PFAS from the air to snow, including the post depositional processes and the assessment of the role of sea-salt aerosol. Therefore, I think that this manuscript needs major revision before it can be accepted.

1. Page 4, line 5. The assessment of blanks is weak. Blanks were performed for the cartridges, but this only covers the potential contamination during analysis in the laboratory. There is no assessment of field blanks, as samples could be contaminated during sampling, handling, transport, and sectioning. This needs some comments.

Response: We thank the Reviewer for their comment concerning the assessment of blanks. We agree with the Reviewer that an ideal field blank would be able to provide background contamination contributed by the ice core extrusion and sectioning. However, the unique procedure of extruding an ice core does not provide a straightforward approach to capturing blank contamination caused by such sampling. A literature search of other ice core sampling studies confirms that field sampling blanks to encompass the sampling acquisition and handling are not typical (Table 1).

Table 1. Literature Survey on Use of Blanks in Recent Ice Core Sampling

| Location | Ice Core Sampling | Blanks | Analytes | Reference |
|---|---|---|---|---|
| Southeast Greenland | 90 m core | No field blanks; only reagent lab blanks. | $CH_3SO_3^-$, $Cl^-$, $SO_4^{2-}$, $NO_3^-$, $Na^+$, $Ca^{2+}$, $NH_4^+$, $Mg^{2+}$, and $K^+$ | (Iizuka et al., 2018) |
| Svalbard, Norway | 5 m core using a Kovacs drill | No field blanks; only reagent lab blanks. | Amino acids and chlorophyll A | (Barbaro et al., 2017) |
| Siberian Altai | 139 m core | No field blanks; only reagent lab blanks | Mercury | (Eyrikh et al., 2017) |
| Svalbard, Norway | 125 m core using a PICO drill | No field blanks; only reagent lab blanks | Elemental carbon | (Ruppel et al., 2017) |
| Akademii Nauk ice core (Siberia) | 129 m | No field blanks; only reagent lab blanks | Aromatic acids (vanillic and para-hydroxybenzoic acids) | (Grieman et al., 2017) |
| Greenland | 87 m and 213 m | No field blanks; only reagent lab blanks | Methanesulfonate, Br, Cl, Na, Ca, S, Ce, and Pb | (Maselli et al., 2017) |
| European Alps | 10 m | No field blanks; | Polybrominated | (Kirchgeorg et |

| | | only reagent lab blanks for sample extraction | diphenyl ethers, perfluorinated acids, polyaromatic hydrocarbons | al., 2016) |
|---|---|---|---|---|
| East Antarctica | 1196 m core | No field blanks; only reagent lab blanks | Organochlorine pesticides | (Bigot et al., 2016) |
| Eastern Switzerland | 101 m using electromechanical-thermal drill | No field blanks; only instrument blanks | Polychlorinated biphenyls | (Pavlova et al., 2015) |
| Tibetan Plateau | 22 m | No field blanks; only reagent blanks for sample extraction | Perfluorinated acids | (Wang et al., 2014) |

We have amassed an annual data set on field blanks (HPLC grade water) transported and exposed to the atmosphere in the Arctic location Resolute Bay, Nunavut since 2010. These samples are shipped back to the lab, extracted and analyzed with methods analogous to the ice samples and compared to the same HPLC grade water stored in the lab. These results indicate that the environmental exposure and shipping do not contribute to PFAS background contamination. We have included tables of this data in the SI (Table S4a-S4b) and added to the main text:

"**Previous results from field blanks (HPLC grade water) transported and exposed to the atmosphere in the Canadian Arctic (Resolute Bay, Nunavut) have indicated that the environmental exposure and shipping do not contribute to background PFAS contamination. We have amassed an annual data set on these field blanks (Table S4a-S4b).**" (P4, L4-L6, Tables S4a-S4b).

We also have some key observations that suggest PFAS background contamination is not compromising our results in this study. We have added text to the manuscript to highlight the measures taken to avoid PFAS contamination such as outfitting the sampling team in PFAS-free apparel, not using polytetrafluoroethene or any other fluorinated polymers during sampling, shipping and handling, rinsing and cleaning all sampling and sample handling equipment with methanol. We've also specified that the Kovacs ice coring drill bit is stainless steel for those unfamiliar with ice coring. For PFAS analysis, the most commonly reported blank contaminant is PFOA arising from polytetrafluoroethene (PFTE) polymers. Our analysis of PFOA in the samples have concentration that are much lower than other ice core analyses suggesting that background contamination was minimal. Any systematic bias encountered through background contamination would not impact the temporal trends.

The text we added to the manuscript now reads:
"**Extensive measures were taken to avoid PFAS contamination during both sample collection and sectioning (i.e. PFAS-free apparel and equipment, methanol-rinsed sampling tools and equipment).**" (P3, L4-L6)

2. Page 5, line 13. As I understand the back trajectories provided by Hysplit, there is an increasing uncertainty for longer time periods. While the 48 hours back trajectories are reliable, there is a huge uncertainty for 10 day back trajectories. I suggest to shorten the back trajectories and discuss the uncertainties.

Response: We thank the Reviewer for this important comment regarding back trajectory analysis uncertainty. It is standard to run 10-day back trajectories when subsequently running a residence time analysis, which greatly reduces the uncertainty in any one pathway (see (Criscitiello et al., 2016) and references therein). If the back trajectory run-time is shortened prior to residence time analysis, this often disregards the atmospheric residence time of the species of interest, resulting in erroneous (incomplete) scaled air mass transport densities that cluster around the trajectory endpoint (ice core site, in this case). In numerous earth science applications, 10-day back trajectories are the standard. For example; (Harris and Kahl, 1994; Huang et al., 2009; Kulshrestha and Kumar, 2014).

In addition to current peer-review literature, we base our methods on discussions contained in the following two books: (1) "Lagrangian Modeling of the Atmosphere" (Lin et al., 2013), which describes in detail why a back trajectory length of 10 days is ideal for such applications, and (2) "Intercontinental Transport of Air Pollution" (Stohl, 2004), which discusses specifically why 10-day back trajectories are used when investigating long-range transport of pollutants to the Arctic.

3. Page 6, line 5. Report and discuss the method for dating the subsamples from the ice core and determining the annual snow accumulation.

Response: We thank the Reviewer for this suggestion. We have indicated in this paragraph where more detail can be found on the ice core dating and have provided a sentence explaining how annual snow accumulation was determined. These sentences read:
"**The dating of the ice core itself is discussed in more detail in the SI (section S1). Annual snow accumulation was determined by measuring the length of the annual ice core sections.** Annual snow accumulation was **calculated as** $0.15 - 0.64$ m yr$^{-1}$…" (P6, L7-L9)

4. Page 6, line 15. "The most: : :." This sentence needs a reference or justification.

Response: We thank the Reviewer for pointing this out. The most straightforward identification of seasonal maxima is in ice cores from the summit of the ice cap. We have added in the reference, (Legrand and Mayewski, 1997). Legrand and Mayewski say that accurate ice dating is based on visual stratigraphy, oxygen isotopic ratios of the ice, and electrical conductivity measurements in areas of high accumulation. The sentence now reads:
"The most **straightforward** temporal record of atmospheric deposition is obtained at the summit of an ice cap **(Legrand and Mayewski, 1997)**;" (P6, L22-23)

5. Page 6, line 21: : :. Compare as well with concentrations and fluxes from maritime Antarctica (Casal et al. EST 2017).

Response: As per the Reviewer's suggestion, we have added PFAA concentrations from Casal et al. (2017) to Table S13 in the SI (P22) and have referenced Casal et al. (2017) in this paragraph (P6, L27-L28).

6. Section 3.1 I suggest to include one or two figures with the patterns, concentrations or fluxes, and extend the discussion. There are some differences in the patterns reported for snow (fresh and/or accumulated) and it is not clear how the results shown here fit with

previous results. For me, the major contribution of this manuscript is the data set of the measurements, much more important than the modelling, however, the paper does not reflect this.

Response: We thank the Reviewer for this comment. Section 3.1 is meant to just be a short discussion on the PFAA concentrations detected on Devon Ice Cap for this study, whilst comparing these measurements to previous studies, particularly the previous two Devon Ice Cap studies. We believe it is important to include a section discussing what has previously been done/measured and how our current results compare. To respond to this Reviewer's comment, the rest of the sections in the paper (sections 3.3 to 3.7) do indeed discuss our measurement data set, where we include multiple figures illustrating fluxes and patterns, among other topics.

7. Page 7, line 10. Provide information for the other years.

Response: As described above and in Section 3.2, we calculated back trajectories for only 2 years to determine whether the unusual PFAA observations of 2013 were caused by a transport anomaly. Preliminary back trajectory analyses that we conducted showed minimal variation from year to year. The year 1994, modeled in Figure 1, was our typical year for comparison, which we used as our baseline, compared to the Meyer et al., 2012 paper. The back trajectory analyses between both their study and our study showed very similar results. We have added in text to section 3.2 to explicitly say this. It now reads:
"The year 1994 was selected to compare to previous trajectory analyses **conducted by Meyer et al, and showed similar results**." (P7, L10-L11)

8. Figure 1 and page 7. I don't agree with the discussion and conclusions for the results shown in figure 1. There is a very weak evidence for source regions in part of Asia, first, due to the uncertainty of the analysis, and secondly, because the signaled regions have a very low population. Figure 1 shows that the Arctic ocean (ice covered or not) may be the main source regions. I wonder if specific ice-influenced photochemistry may affect the formation of the targeted compounds.

Response: It is not clear what type of ice-mediated photochemistry the Reviewer is referring to. Perhaps the Reviewer is referring to enhanced PFAA formation from precursors? There is currently no evidence in literature for ice-mediated PFAS formation to warrant speculative discussions. However, the most thoroughly studied mechanisms for producing PFAA from atmospheric precursors, are presented in the discussion with supporting data to postulate sources of PFAAs in this remote ice cap including the measured temporal deposition profiles, known production inventories, air mass transport density analysis, PFAA molar flux ratios of homologue pairs, major ion tracers, and previous modeling studies on air mass transport (see response to comment 2). We have added a sentence to section 3.7 to warrant a focus on this subject for future work:
"**The role of ice in the formation of PFAAs from precursors is currently unknown and future work should focus on ice-mediated PFAA formation**." (P18, L3-L4)

9. I would suggest to try to correlate the concentrations and fluxes with the extension of the arctic ice cap.

We thank the Reviewer for this innovative suggestion. We have correlated the annual PFAS deposition fluxes with minimum Arctic sea ice extent and sea ice area. The results of the correlations are now mentioned in the text and reported in the SI:

"Additionally, correlations between annual PFAS deposition and Arctic sea ice minimum were calculated (NSIDC, 2017) (Table S22). If marine aerosols were a major source of PFAS to the ice cap, then a negative correlation between annual sea ice extent and deposition would be expected, but the majority (12 of 14) showed a positive correlation." (P16, L22-L25)

**Table S22. Coefficients of determination ($R^2$) and statistical significance (p) between PFAS deposition and Arctic sea ice extent and area. Sea ice time series are based on SMMR/SMM/I satellite observations. The slope sign is indicated as either positive (+) or negative (-) for each PFAA compound. Correlations are ranked in terms of significance by purple>blue>orange.**

| | Sea Ice Extent | Sea Ice Area | Slope Sign (+/-) |
|---|---|---|---|
| **PFPeA** | $R^2$= 0.047 p=0.2088 | $R^2$= 0.050 p=0.1959 | + |
| **PFHxA** | $R^2$= 0.217 p=0.0048 | $R^2$= 0.215 p=0.0050 | + |
| **PFHpA** | $R^2$= 0.295 p=0.0007 | $R^2$= 0.295 p=0.0007 | + |
| **PFOA** | $R^2$= 0.106 p=0.0559 | $R^2$= 0.114 p=0.0470 | + |
| **PFNA** | $R^2$= 0.183 p=0.0104 | $R^2$= 0.199 p=0.0073 | + |
| **PFDA** | $R^2$= 0.129 p=0.0339 | $R^2$= 0.142 p=0.0258 | + |
| **PFUnDA** | $R^2$= 0.161 p=0.0168 | $R^2$= 0.169 p=0.0140 | + |
| **PFDoDA** | $R^2$= 0.058 p=0.1648 | $R^2$= 0.067 p=0.1334 | + |
| **PFTrDA** | $R^2$= 0.030 p=0.3172 | $R^2$= 0.033 p=0.2938 | + |
| **PFOcDA** | $R^2$= 0.007 p=0.6437 | $R^2$= 0.017 p=0.4618 | - |
| **PFBS** | $R^2$= 0.113 p=0.0485 | $R^2$= 0.116 p=0.0451 | + |
| **PFHpS** | $R^2$= 0.055 p=0.1735 | $R^2$= 0.049 p=0.2000 | + |
| **PFOS** | $R^2$= 0.099 p=0.0659 | $R^2$= 0.085 p=0.0887 | + |
| **FOSA** | $R^2$= 0.266 p=0.0015 | $R^2$= 0.261 p=0.0017 | - |

10. Page 13, lines 4-5. I don't understand this sentence.

Response: We thank the Reviewer for mentioning this. We have clarified this sentence, which now reads:

"Indirect sources of PFAAs are contributors to the global presence of these compounds, **particularly in locations such as the Canadian Archipelago** (Benskin et al., 2012a)." (P14, L4-L5)

11. An implicit assumption in this work is that the observed pattern and concentrations are a direct consequence of atmospheric snow deposition, thus snow scavenging of atmospheric PFAS. No discussion is made of post-depositional processes affecting the concentrations and patterns of PFAS.

Response: In regards to the post-depositional processes, we do discuss this in some detail in section 3.3 (P8, L4-L24). Melting events can occur in the ice core, which can cause PFAAs to elute into the snowpack and refreeze at ice layer interfaces. MacInnis et al. (2017) discuss these post-depositional processes in more detail. To make this discussion more implicit, we have added more detail on these post-depositional processes to this paragraph, which now reads: "One study has examined elution behaviour of PFAAs from a melting snowpack and found that elution of PFAAs is driven by **a number of factors including water solubility in the snowpack, PFAA chain length, and sorptive capacity of the snow grain surface** (Plassmann et al., 2011). **Due to melting at the surface, concentrations of PFAAs measured near the surface layer can result in inaccurate estimations whereby one year might be overestimated and another year underestimated. Melting events on the ice cap have** likely happened periodically over the last 15 – 20 years, thereby blurring to some extent, the vertical profile. However, we expect any melting that occurred to have primarily affected the seasonal trends**, as Koerner, 2005 states that the percolation of meltwater in a snowpack will refreeze within an annual layer, and so seasonal cycles of PFAA deposition will be biased, but annual interpretations should not be affected (Koerner, 2005)**." (P8, L12-L20)

12. Even If I was convinced that most PFAS in the studied arctic region come from atmospheric oxidation of neutral precursors, no discussion is made on the processes responsible for the transport of ionizable PFAS from the gas phase (I guess oxidation occurs in the gas phase) to deposited snow. Once a ionizable PFAS is formed, it may attach rapidly to aerosols. Which are the dominant aerosol types in this region? A reference is needed to support the response to this question. Furthermore, it could be that neutral precursors are the main contributors to ionizable PFAS in the surface Arctic ocean, and then these are transferred to Devon Cap by sea-salt aerosol formation and deposition.

Response: It is not clear what the Reviewer is referring to by "aerosol type" – size, inorganic composition, organic composition, nucleating ability, optical depth, water content? We feel the scope of identifying aerosol types in the Devon Ice Cap atmosphere is beyond the scope of this study. However, what is directly relevant to this paper is the differences in physical properties of the precursors (neutral and volatile) and the acids (ionic and sorptive). As we mention in the introduction, after PFAAs are formed in the gas phase, they can be deposited to the surface through both wet and dry deposition due to their physicochemical properties. We have clarified our introductory sentence and added references to a few modelling studies:

"Once these PFAAs are formed in the **gas phase**, they undergo wet or dry deposition **to the surface. The specifics of these processes have been considered in several modelling studies**

in the literature (Armitage et al., 2009a, 2009b; Schenker et al., 2008; Wallington et al., 2006; Yarwood et al., 2007)." (P2, L18-L21)

13. Page 15, line 5. I don't know any study on the occurrence of PFAS in the marine surface microlayer, but for other POPs, there is a huge variability on the enrichment factors, thus a lack of correlation does not contradict the potential role of marine aerosols. Furthermore, Na can be forced to move out from the snow/ice after deposition.

Response: The mechanisms for sea surface microlayer enrichment of PFAS are not analogous to other POPs due to the unique properties imparted by the combination of the hydrophobic and lipophobic perfluoroalkyl chain and the hydrophilic carboxylate or sulfonate moiety. A preliminary study of enrichment of PFAS in the freshwater surface microlayer has been described (Reth et al., 2011), with enrichment increasing with chain length. As discussed in a previous paper from our team (and briefly mentioned in this work), surfactant strength does not describe the differences between PFAS levels in the ice cap and ocean. To clarify this, we have added a reference to that paper at the beginning of the discussion of ice cap/ocean comparisons to make clear we are using a previously described data analysis technique:
"Another technique used to assess the influence of marine aerosol deposition of PFAAs to the Devon Ice Cap is a comparison between ocean and ice cap homologue patterns **(MacInnis et al., 2017)**." (P16, L7-L8)
We have further clarified by re-phrasing the description of the surfactant relationship:
"**Water-to-air transport of PFAS is related to surfactant strength (Reth et al., 2011). Consistent with results from MacInnis et al. (2017), differences in surfactant strength cannot account for the different homologue profiles observed on the ice cap and in the ocean.**" (P16, L10-L12)

We agree with the Reviewer that post-depositional processes may be different for PFAS and sodium, although this has not been explicitly studied. On its own, these correlations would not provide sufficient evidence to discount the role of marine aerosols. However, the combination of the lack of correlation with sodium and sea ice extent, and the differing homologue patterns support our conclusion that marine aerosols are not the dominant source of PFAS to Devon Ice Cap.

14. Page 15, line 9. This is not true because the patterns in snow are different than in seawater even for a given site, and because we do not know the patterns in the surface microlayer, nor in remote aerosols.

Response: As described above in the response to comment 13, we do have information on the patterns of enrichment for the surface microlayer and marine aerosols relative to ocean water (Reth et al., 2011). To clarify this point, we have moved our discussion of water-to-air transport to directly follow this sentence. The section now reads as follows:
"If marine aerosols were/are a major source of PFAA contamination on Devon Ice Cap, then one would expect the homologue profiles to be similar between the ocean and the ice cap. **Water-to-air transport of PFAS is related to surfactant strength (Reth et al., 2011). Consistent with results from MacInnis et al. (2017), differences in surfactant strength cannot account for the different homologue profiles observed on the ice cap and in the ocean (Fig S8)**." (P16, L8-L12)

15. I suggest to plot the ratio of concentrations between Devon cap and arctic ocean (except for PFBA which has clearly a different origin and behavior), and plot this ratio versus the number of C of the pfas chain, and then discuss taking into account the literature.

Response: We thank the Reviewer for this suggestion. We actually initially plotted the ratios of concentrations in the Devon Ice Cap compared to ocean levels, similar to what was done in MacInnis et al. (2017), but included a proportional analysis of the molar concentrations in this paper instead, as to avoid duplicating a previous paper's figure. We have added Figure S8 to the SI (P31-P32)) and included a reference to it in the main text:
"Consistent with results from MacInnis et al. (2017), differences in surfactant strength cannot account for the different homologue profiles observed on the ice cap and in the ocean **(Fig. S8).**"
(P16, L10-L12)

[Figure]

**Figure S8. Ratios of observed concentrations in the Devon Ice Cap compared to levels in the Arctic Ocean (Benskin et al., 2012b)(Cai et al., 2012), Canadian Artic Archipelago (Benskin et al., 2012b), Greenland Sea (Zhao et al., 2012)(Busch et al., 2010), North Sea (Ahrens et al., 2010b), North Atlantic Ocean (Ahrens et al., 2010a)(Ahrens et al., 2009)(Zhao et al., 2012), and North Pacific Ocean (Cai et al., 2012) in 2005 and 2007 to 2010.**

16. The authors insist that sea salt aerosol does not play a role, but I don't see concluding arguments. Which is the main source of aerosols in the region and how ionizable pfas behave after their formation? Ok, let's assume that sea-salt aerosols do not play an important role, then, to which aerosols are PFAS bound to? Are they in the gas phase and then scavenged by snow? Please, provide a plausible mechanistic explanation.

Response: As we have described above, PFAS formed in the gas phase will be deposited through wet and dry deposition (see response to comment 12). The identity of the aerosols with which PFAS might be associated is an open question. Our correlations with different tracers (e.g. for dust) were targeted to reduce uncertainty in this area.

17. Pages 16-17. A hypothetical dust source is commented, but this is not supported by the assessment of air mass back trajectories.

Response: Our discussion regarding dust is not based on our back trajectories, but rather on the well-established literature indicating both the presence of dust in the Arctic and that it can be coming from LRT (Darby et al., 1974; Rahn et al., 1977; Zdanowicz et al., 1998). To clarify this in the text, we have rephrased and added an additional reference:

"Mineral dust aerosols can accumulate acidic atmospheric contaminants, such as nitric and hydrochloric acids (Sullivan et al., 2007). **Dust can undergo LRT and deposit in the remote Arctic** (Rahn et al., 1977; **Zdanowicz et al., 1998**)." (P18, L6-L8)

18. Note that Ca and Mg are enriched in sea-salt aerosol coming from the sea surface microlayer (Jayarathne et al. EST 2016)! Then the correlations between Ca and PFAS would support sea salt as an important contributor to PFAS at the studied site!

Response: We thank the Reviewer for mentioning this. Although we did consider the enrichment of calcium and magnesium, we believe it has a negligible impact on our interpretation of the data. Enrichment in marine aerosols by calcium has been shown by a few studies (Jayarathne et al., 2016; Oppo et al., 1999; Salter et al., 2016). Aerosol enrichment by magnesium has been observed in a single study (Jayarathne et al., 2016), while another showed no enrichment (Oppo et al., 1999). Using established seawater cation ratios, we calculated average non-sea-salt (nss) fractions of calcium and magnesium of 99.5 % and 70.3 %, respectively (Table S19). Jayarathne et al. (2016) observed calcium enrichment factors were variable, but up to 3.38 in marine aerosols. If we assumed this maximum enrichment, nss-Ca on the ice cap would remain >98 %. It is extremely unlikely that this small change would impact the interpretation of our correlations. Given the uncertainty and observed variability in enrichment for magnesium, it is not possible at this time to determine the impact of possible magnesium marine aerosol enrichment on ice cap nss-Mg.

19. After reading this manuscript I think that the interpretation needs to be re-evaluated, and a new version prepared taking into account my comments above.

Response: We thank the Reviewer for taking the time to read our manuscript and provide important feedback. We have addressed all of the reviewer's comments and prepared a revised manuscript.

**References**

Ahrens, L., Barber, J. L., Xie, Z. and Ebinghaus, R.: Longitudinal and Latitudinal Distribution of Perfluoroalkyl Compounds in the Surface Water of the Atlantic Ocean, Environ. Sci. Technol., 43(9), 3122–3127, doi:10.1021/es803507p, 2009.

Ahrens, L., Xie, Z. and Ebinghaus, R.: Distribution of perfluoroalkyl compounds in seawater from Northern Europe, Atlantic Ocean, and Southern Ocean, Chemosphere, 78(8), 1011–1016, doi:10.1016/j.chemosphere.2009.11.038, 2010a.

Ahrens, L., Gerwinski, W., Theobald, N. and Ebinghaus, R.: Sources of polyfluoroalkyl compounds in the North Sea, Baltic Sea and Norwegian Sea: Evidence from their spatial distribution in surface water, Mar. Pollut. Bull., 60(2), 255–260, doi:10.1016/j.marpolbul.2009.09.013, 2010b.

Armitage, J. M., Macleod, M. and Cousins, I. T.: Comparative assessment of the global fate and transport pathways of long-chain perfluorocarboxylic acids (PFCAs) and perfluorocarboxylates (PFCs) emitted from direct sources, Environ. Sci. Technol., 43(15), 5830–5836, 2009a.

Armitage, J. M., Schenker, U., Martin, J. W., Macleod, M. and Cousins, I. T.: Modeling the global fate and transport of perfluorooctane sulfonate (PFOS) and precursor compounds in relation to temporal trends in wildlife exposure, Environ. Sci. Technol., 43, 9274–9280, 2009b.

Barbaro, E., Spolaor, A., Karroca, O., Park, K.-T., Martma, T., Isaksson, E., Kohler, J., Gallet, J. C., Bjorkman, M. P., Cappelletti, D., Spreen, G., Zangrando, R., Barbante, C. and Gambaro, A.: Free amino acids in the Arctic snow and ice core samples: Potential markers for paleoclimatic studies, Sci. Total Environ., 607–608, 454–462, doi:10.1016/j.scitotenv.2017.07.041, 2017.

Benskin, J. P., Ahrens, L., Muir, D. C. G., Scott, B. F., Spencer, C., Rosenberg, B., Tomy, G., Kylin, H., Lohmann, R. and Martin, J. W.: Manufacturing origin of perfluorooctanoate (PFOA) in Atlantic and Canadian Arctic seawater, Environ. Sci. Technol., 46, 677–685, 2012a.

Benskin, J. P., Muir, D. C. G., Scott, B. F., Spencer, C., De Silva, A. O., Kylin, H., Martin, J. W., Morris, A., Lohmann, R., Tomy, G., Rosenberg, B., Taniyasu, S. and Yamashita, N.: Perfluoroalkyl Acids in the Atlantic and Canadian Arctic Oceans, Environ. Sci. Technol., 46(11), 5815–5823, doi:10.1021/es300578x, 2012b.

Bigot, M., Curran, M. A. J., Moy, A. D., Muir, D. C. G., Hawker, D. W., Cropp, R., Teixeira, C. F. and Bengtson Nash, S. M.: Brief communication: Organochlorine pesticides in an archived firn core from Law Dome, East Antarctica, Cryosph., 10(5), 2533–2539, doi:10.5194/tc-10-2533-2016, 2016.

Busch, J., Ahrens, L., Xie, Z., Sturm, R. and Ebinghaus, R.: Polyfluoroalkyl compounds in the East Greenland Arctic Ocean, J. Environ. Monit., 12(6), 1242, doi:10.1039/c002242j, 2010.

Cai, M., Zhao, Z., Yin, Z., Ahrens, L., Huang, P., Cai, M., Yang, H., He, J., Sturm, R., Ebinghaus, R. and Xie, Z.: Occurrence of Perfluoroalkyl Compounds in Surface Waters from the North Pacific to the Arctic Ocean, Environ. Sci. Technol., 46(2), 661–668, doi:10.1021/es2026278, 2012.

Criscitiello, A. S., Marshall, S. J., Evans, M. J., Kinnard, C., Norman, A.-L. and Sharp, M. J.: Marine aerosol source regions to Prince of Wales Icefield, Ellesmere Island, and influence from the tropical Pacific, 1979-2001, J. Geophys. Res. Atmos., 121(16), 9492–9507, doi:10.1002/2015JD024457, 2016.

Darby, D., Burckle, L. and Clarck, D.: Airborne dust on the Arctic pack ice, its composition and fallout rate, Earth Planet. Sci. Lett., 24, 166–172, 1974.

Eyrikh, S., Eichler, A., Tobler, L., Malygina, N., Papina, T. and Schwikowski, M.: A 320 Year Ice-Core Record of Atmospheric Hg Pollution in the Altai, Central Asia, Environ. Sci. Technol., 51(20), 11597–11606, doi:10.1021/acs.est.7b03140, 2017.

Grieman, M. M., Aydin, M., Fritzsche, D., McConnell, J. R., Opel, T., Sigl, M. and Saltzman, E. S.: Aromatic acids in a Eurasian Arctic ice core: a 3000-year proxy record of biomass burning, Clim. Past Discuss., 1–29, doi:10.5194/cp-2016-126, 2017.

Harris, J. M. and Kahl, J. D. W.: Analysis of 10-day isentropic flow patterns for Barrow, Alaska: 1985–1992, J. Geophys. Res., 99(D12), 25845–25855, doi:10.1029/94JD02324, 1994.

Huang, J., Zhang, C. and Prospero, J. M.: Large-scale effect of aerosols on precipitation in the West African Monsoon region, Q. J. R. Meteorol. Soc., 135(640), 581–594, doi:10.1002/qj.391, 2009.

Iizuka, Y., Uemura, R., Fujita, K., Hattori, S., Seki, O., Miyamoto, C., Suzuki, T., Yoshida, N., Motoyama, H. and Matoba, S.: A 60 Year Record of Atmospheric Aerosol Depositions Preserved in a High-Accumulation Dome Ice Core, Southeast Greenland, J. Geophys. Res. Atmos., 123(1), 574–589, doi:10.1002/2017JD026733, 2018.

Jayarathne, T., Sultana, C. M., Lee, C., Malfatti, F., Cox, J. L., Pendergraft, M. A., Moore, K. A., Azam, F., Tivanski, A. V, Cappa, C. D., Bertram, T. H., Grassian, V. H., Prather, K. A. and Stone, E. A.: Enrichment of Saccharides and Divalent Cations in Sea Spray Aerosol During Two Phytoplankton Blooms, Environ. Sci. Technol., 50(21), 11511–11520, doi:10.1021/acs.est.6b02988, 2016.

Kirchgeorg, T., Dreyer, A., Gabrielli, P., Gabrieli, J., Thompson, L. G., Barbante, C. and Ebinghaus, R.: Seasonal accumulation of persistent organic pollutants on a high altitude glacier in the Eastern Alps, Environ. Pollut., 218, 804–812, doi:10.1016/j.envpol.2016.08.004, 2016.

Koerner, R. M.: Mass balance of glaciers in the Queen Elizabeth Islands, Nunavut, Canada, Ann. Glaciol., 42(Table 1), 417–423, doi:10.3189/172756405781813122, 2005.

Kulshrestha, U. and Kumar, B.: Airmass Trajectories and Long Range Transport of Pollutants: Review of Wet Deposition Scenario in South Asia, Adv. Meteorol., 2014, 1–14, doi:10.1155/2014/596041, 2014.

Legrand, M. and Mayewski, P.: Glaciochemistry of polar ice cores: A review, Rev. Geophys., 35(3), 219–243, doi:10.1029/96RG03527, 1997.

Lin, J., Brunner, D., Gerbig, C., Stohl, A., Luhar, A. and Webley, P.: Lagrangian Modeling of the Atmosphere, American Geophysical Union., 2013.

MacInnis, J. J., French, K., Muir, D. C. G., Spencer, C., Criscitiello, A., De Silva, A. O. and Young, C. J.: A 14-year depositional ice record of perfluoroalkyl acids in the High Arctic, Environ. Sci. Process. Impacts, 19, 22–30, doi:10.1039/C6EM00593D, 2017.

Maselli, O. J., Chellman, N. J., Grieman, M., Layman, L., McConnell, J. R., Pasteris, D., Rhodes, R. H., Saltzman, E. and Sigl, M.: Sea ice and pollution-modulated changes in Greenland ice core methanesulfonate and bromine, Clim. Past, 13(1), 39–59, doi:10.5194/cp-13-39-2017, 2017.

NSIDC: Arctic Sea Ice Minimum, National Snow and Ice Data Center, [online] Available from: https://climate.nasa.gov/vital-signs/arctic-sea-ice/ (Accessed 1 March 2018), 2017.

Oppo, C., Bellandi, S., Degli Innocenti, N., Stortini, A. M., Loglio, G., Schiavuta, E. and Cini, R.: Surfactant components of marine organic matter as agents for biogeochemical fractionation and pollutant transport via marine aerosols, Mar. Chem., 63(3–4), 235–253, doi:10.1016/S0304-4203(98)00065-6, 1999.

Pavlova, P. A., Jenk, T. M., Schmid, P., Bogdal, C., Steinlin, C. and Schwikowski, M.: Polychlorinated Biphenyls in a Temperate Alpine Glacier: 1. Effect of Percolating Meltwater on their Distribution in Glacier Ice, Environ. Sci. Technol., 49(24), 14085–14091, doi:10.1021/acs.est.5b03303, 2015.

Plassmann, M. M., Meyer, T., Lei, Y. D., Wania, F., McLachlan, M. S. and Berger, U.: Laboratory Studies on the Fate of Perfluoroalkyl Carboxylates and Sulfonates during Snowmelt, Environ. Sci. Technol., 45(16), 6872–6878, doi:10.1021/es201249d, 2011.

Rahn, K. A., Borys, R. D. and Shaw, G. E.: The Asian source of Arctic haze bands, Nature, 268(5622), 713–715, doi:10.1038/268713a0, 1977.

Reth, M., Berger, U., Broman, D., Cousins, I. T., Nilsson, E. D. and McLachlan, M. S.: Water-to-air transfer of perfluorinated carboxylates and sulfonates in a sea spray simulator, Environ. Chem., 8, 381–388, 2011.

Ruppel, M. M., Soares, J., Gallet, J.-C., Isaksson, E., Martma, T., Svensson, J., Kohler, J., Pedersen, C. A., Manninen, S., Korhola, A. and Ström, J.: Do contemporary (1980–2015) emissions determine the elemental carbon deposition trend at Holtedahlfonna glacier, Svalbard?, Atmos. Chem. Phys., 17(20), 12779–12795, doi:10.5194/acp-17-12779-2017, 2017.

Salter, M. E., Hamacher-Barth, E., Leck, C., Werner, J., Johnson, C. M., Riipinen, I., Nilsson, E. D. and Zieger, P.: Calcium enrichment in sea spray aerosol particles, Geophys. Res. Lett., 43(15), 8277–8285, doi:10.1002/2016GL070275, 2016.

Schenker, U., Scheringer, M., MacLeod, M., Martin, J. W., Cousins, I. T. and Hungerbühler, K.: Contribution of Volatile Precursor Substances to the Flux of Perfluorooctanoate to the Arctic, Environ. Sci. Technol., 42(10), 3710–3716, doi:10.1021/es703165m, 2008.

Stohl, A.: Intercontinental Transport of Air Pollution, Springer-Verlag Berlin Heidelberg., 2004.

Sullivan, R. C., Guazzotti, S. A., Sodeman, D. A. and Prather, K. A.: Direct observations of the atmospheric processing of Asian mineral dust, Atmos. Chem. Phys., 7, 1213–1236, doi:10.5194/acpd-6-4109-2006, 2007.

Wallington, T. J., Hurley, M. D., Xia, J., Wuebbles, D. J., Sillman, S., Ito, A., Penner, J. E., Ellis, D. A., Martin, J., Mabury, S. A., Nielsen, O. J. and Sulbaek Andersen, M. P.: Formation of C7F15COOH (PFOA) and Other Perfluorocarboxylic Acids during the Atmospheric Oxidation of 8:2 Fluorotelomer Alcohol, Environ. Sci. Technol., 40(3), 924–930, doi:10.1021/es051858x, 2006.

Wang, X., Halsall, C., Codling, G., Xie, Z., Xu, B., Zhao, Z., Xue, Y., Ebinghaus, R. and Jones, K. C.: Accumulation of Perfluoroalkyl Compounds in Tibetan Mountain Snow: Temporal Patterns from 1980 to 2010, Environ. Sci. Technol., 48(1), 173–181, doi:10.1021/es4044775, 2014.

Yarwood, G., Kemball-Cook, S., Keinath, M., Waterland, R. L., Korzeniowski, S. H., Buck, R. C., Russell, M. H. and Washburn, S. T.: High-resolution atmospheric modeling of fluorotelomer alcohols and perfluorocarboxylic acids in the North American troposphere, Environ. Sci. Technol., 41, 5756–5762, 2007.

Zdanowicz, C. M., Zielinski, G. A. and Wake, C. P.: Characteristics of modern atmospheric dust deposition in snow on the Penny Ice Cap, Baffin Island, Arctic Canada, Tellus B, 50(5), 506–520, doi:10.1034/j.1600-0889.1998.t01-1-00008.x, 1998.

Zhao, Z., Xie, Z., Möller, A., Sturm, R., Tang, J., Zhang, G. and Ebinghaus, R.: Distribution and long-range transport of polyfluoroalkyl substances in the Arctic, Atlantic Ocean and Antarctic coast, Environ. Pollut., 170, 71–77, doi:10.1016/j.envpol.2012.06.004, 2012.